


# Isogeometric analysis of the lithosphere under topographic loading: Igalith v1.0.0

Rozan Rosandi[1], Yudi Rosandi[2], and Bernd Simeon[1]

[1]RPTU Kaiserslautern-Landau
[2]Universitas Padjadjaran

**Correspondence:** Rozan Rosandi (rozan.rosandi@math.rptu.de)

**Abstract.** This paper presents methods from isogeometric finite element analysis for numerically solving problems in geoscience involving partial differential equations. In particular, we consider the numerical simulation of shells and plates in the context of isostasy. Earth's lithosphere is modeled as a thin elastic shell or plate floating on the asthenosphere and subject to topographic loads. We demonstrate the computational methods on the isostatic boundary value problem posed on selected geographic locations. For Europe, the computed lithospheric depression is compared with available Mohorovičić depth data. We also perform parameter identification for the effective elastic thickness of the lithosphere, the rock density, and the topographic load that are most plausible to explain the measured depths. An example of simulating the entire lithosphere of the Earth as a spherical shell using multi-patch isogeometric analysis is presented, which provides an alternative to spherical harmonics for solving partial differential equations on a spherical domain.

## 1 Introduction

Finite element methods have been widely used to compute numerical approximations of solutions to partial differential equations. In standard finite element methods, the computational domain is subdivided into parts that are images of elementary geometric shapes, called finite elements, on which a number of shape functions are defined. Usually, the shape functions are polynomial functions determined by interpolation conditions on some reference element. Joining together all the elements along with the shape functions yields a finite element space in which a numerical solution to the problem is sought. It is constructed by finding a linear combination of the shape functions on each element that best approximates the exact solution (Brenner and Scott, 1994; Braess, 2007; Zienkiewicz et al., 2013).

Globally $C^1$ finite element spaces are required for a conforming discretization of higher-order problems, such as the shell and plate problems considered in this work. The construction of such spaces is generally computationally expensive and requires a lot of degrees of freedom per element. This difficulty has led to various methods for solving the shell and plate equations more efficiently. An example is the non-conforming mixed formulation given by the classical discrete Kirchhoff triangular (DKT) elements (Batoz et al., 1980), where the $C^1$ condition is imposed only at the nodes of the mesh. Other examples include the use of rotation-free (RF) elements (Oñate and Zárate, 2000), assumed natural deviatoric strain (ANDES) elements (Mostafa et al., 2013), discontinuous Galerkin (DG) methods (Engel et al., 2002), and the Hellan–Herrmann–Johnson (HHJ) method



(Neunteufel and Schöberl, 2019). Another way to address the problem is to apply isogeometric finite element methods (Kiendl et al., 2009), which is the main topic of this work.

Isogeometric analysis (IGA) is a computational paradigm for solving partial differential equations (PDE) that employs the same shape functions used to describe the domain of the problem to construct finite element approximations of solutions to the problem. It allows the integration of finite element analysis (FEA) with technologies from computer-aided design (CAD).

The concept of isogeometric analysis is first presented in the seminal work by Hughes et al. (2005). Standard references on the subject include Cottrell et al. (2009), Buffa and Sangalli (2016), Lyche et al. (2018), Jüttler and Simeon (2015), and van Brummelen et al. (2021).

B-splines and non-uniform rational B-splines (NURBS) are conventionally used for the shape functions in isogeometric analysis. One advantage of using them for shell and plate problems is the simple construction of $C^1$ isogeometric spline spaces

on a single patch to discretize the equations with less degrees of freedom than standard $C^1$ finite element methods. Further features and capabilities of isogeometric analysis presented in this paper are the exact representation of curved domains, the coupling of multiple patches, which preserves the $C^1$ continuity along the interfaces (Kapl et al., 2018; Farahat et al., 2023), and adaptive local refinement using hierarchical B-splines (Vuong et al., 2011; Garau and Vázquez, 2018; Buffa et al., 2022).

We conduct numerical experiments on various geographic locations using the global topography data from Earth2014 (Hirt

and Rexer, 2015). A Mohorovičić depth map is available for the European Plate (Grad et al., 2009), which is used to verify the results. Information about the ground truth additionally allows us to estimate unknown parameters of the model via least-squares methods constrained by the governing equations. This is applied to identify the spatial distribution of the effective elastic thickness, the density of overlying rock, and the topographic load that are most plausible to explain the measured data for the Mohorovičić depth.

We begin with the description of the mathematical models that are used in this work to derive the equilibrium equations for shells and plates in the context of isostasy. Section 3 introduces isogeometric analysis and the methods used to discretize and numerically solve boundary value problems using B-splines and NURBS. In Sect. 4, we provide a method to estimate parameters of the model using available real-world data. Application of the methods to selected geographic locations is discussed in Sect. 5, followed by a summary and conclusions in the last section of the paper.

## 50  2  Mathematical model of the lithosphere

In this work, the term lithosphere refers to the solid part of Earth's interior that responds elastically to applied mechanical loads on time scales of geologic duration. It encompasses Earth's outermost layer, the crust, and a portion of Earth's upper mantle (see Fig. 1). This particular notion is known as the elastic lithosphere and is to be distinguished from other definitions (see Melosh, 2011, Box 3.4). Since the mechanical behavior of a planet's interior depends on the rheology of the material of which

it is composed and the duration of the loads under consideration, the location and size of the lithosphere are rather ill-defined. Nevertheless, the concept of an elastic lithosphere has proven useful for modeling purposes.



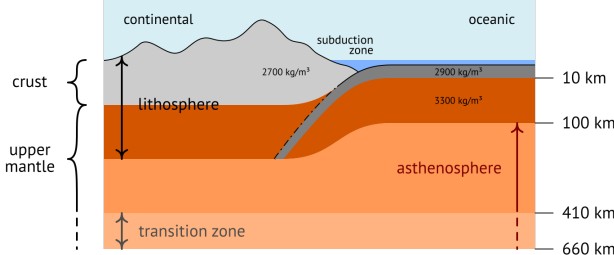

**Figure 1.** Top layers of the Earth (Lowrie, 1997; Rogers, 2008).

We treat the lithosphere as an elastic shell floating on the asthenosphere and subject to gravitational body forces. The asthenosphere comprises the mechanically weak and ductile region of Earth's upper mantle, which behaves like a viscous fluid on geologic time scales and exerts an outward buoyancy force on the lithosphere. The magnitude of the force is proportional to the pressure difference between the fluid and the submerged body. According to Archimedes' principle, it is equal to the weight of the displaced fluid, which, in our case, depends on the depression of the lithosphere. The weight of topography is treated as a gravitational load, i.e., an inward force proportional to topographic elevation and rock density acts on the lithosphere, which causes its depression. Isostasy or isostatic equilibrium refers to the state of mechanical equilibrium between the lithosphere and the asthenosphere due to gravity and buoyancy (Gutenberg, 1949; Watts, 2001).

In the following, we introduce some basic concepts from the theory of elastic shells and plates to formulate a mathematical model for the lithosphere as described above. For a more elaborate introduction to mathematical elasticity and thin shell structures, we refer to Marsden and Hughes (1994) and Bischoff et al. (2004), respectively.

## 2.1 Shell and plate models

A shell is a three-dimensional solid whose thickness in one dimension is considerably small relative to the other two dimensions. The mathematical model of a shell can be reduced to a two-dimensional one by considering only the mechanics on some reference surface (see Fig. 2 for an illustration).

One typically distinguishes between thick and thin shell models. Thick shell models capture transverse shear strains in addition to membrane and bending strains, as opposed to thin shell models, where the shell thickness is assumed small enough so that the effects of transverse shear deformations can be neglected. The configuration of a thin shell is fully determined by the position of its reference surface in physical space, whereas the configuration of a thick shell is supplemented by a deformable vector field on the reference surface, called a director field. A surface together with a director field is referred to as a one-director Cosserat surface.

In the case where the initial undeformed configuration of the reference surface is planar and there are no membrane strains, one speaks of a plate instead of a shell. The displacement of the reference surface from the initial configuration is then reduced to its vertical deflection perpendicular to the reference plane.





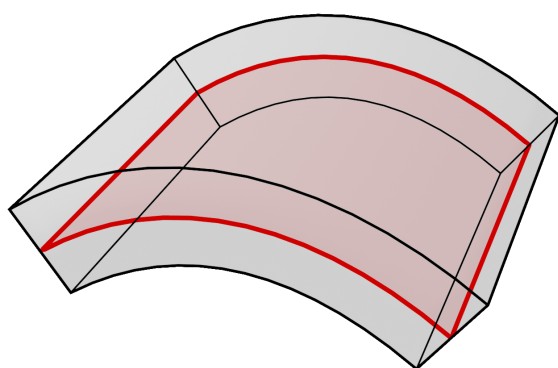

**Figure 2.** A shell segment and its reference surface (red).

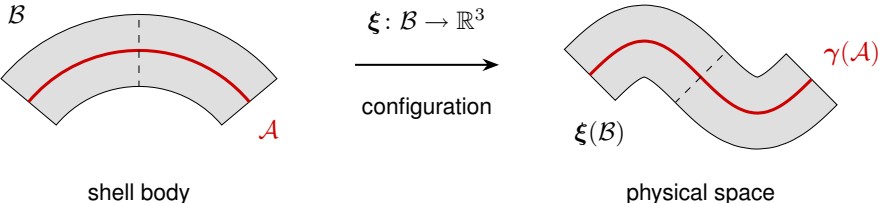

**Figure 3.** Shell configuration, corresponding mid-surface configuration (red line), and a fiber of the shell (dashed line).

To showcase the capabilities of isogeometric analysis in solving higher-order problems numerically, we focus on the displacement formulation of the Koiter model for thin shells and the Kirchhoff–Love model for thin plates, which require $C^1$ finite elements for a conforming discretization. Depending on the ratio of the shell thickness to the scale of the simulation, a thick shell model might be more adequate for capturing the correct behavior of the lithosphere.

### 2.1.1 Shell configuration

Let $\mathcal{B}$ denote the shell body, modeled as a three-dimensional differentiable manifold with boundary, and $\mathcal{A} \subset \mathcal{B}$ its reference surface, chosen to be the middle surface of the shell. A shell consists of fibers that are transverse to the reference surface. A configuration of $\mathcal{B}$ in the physical space $\mathbb{R}^3$ is a mapping $\boldsymbol{\xi} \colon \mathcal{B} \to \mathbb{R}^3$, which assigns a spatial point $\boldsymbol{\xi}(\boldsymbol{X}) \in \mathbb{R}^3$ to each particle $\boldsymbol{X} \in \mathcal{B}$. We restrict ourselves to configurations that are continuously differentiable embeddings of $\mathcal{B}$ in $\mathbb{R}^3$, i.e., $\boldsymbol{\xi}$ is at least a $C^1$ diffeomorphism onto its image. Thus, phenomena such as folding, ripping, pinching, or interpenetration of matter are excluded from our model.

There are various conditions that can be imposed on configurations of a shell. In this work, we consider the following kinematic assumptions:

(A1) Mindlin hypothesis: Fibers are mapped to straight line segments that vary linearly in terms of the thickness parameter,



(A2) Kirchhoff hypothesis: Fibers are mapped to curve segments that are orthogonal to the reference surface,

(A3) Inextensibility: The length of each fiber remains constant for all configurations.

Under the Mindlin hypothesis, any admissible configuration of the shell can be locally represented using curvilinear coordinates $\boldsymbol{\vartheta} = (\vartheta^1, \vartheta^2, \vartheta^3)$ by a mapping of the form

$$\boldsymbol{\xi}(\vartheta^1, \vartheta^2, \vartheta^3) = \boldsymbol{\gamma}(\vartheta^1, \vartheta^2) + \vartheta^3 \boldsymbol{\eta}(\vartheta^1, \vartheta^2), \tag{1}$$

where $\boldsymbol{\gamma} = \boldsymbol{\xi}|_{\mathcal{A}}$ is the mid-surface configuration and $\boldsymbol{\eta}$ is a unit director field, which coincides with the unit normal $\boldsymbol{n}$ to the mid-surface if the Kirchhoff hypothesis is assumed. The thickness parameter $\zeta = \vartheta^3$ ranges from a lower bound $t^-$ to an upper bound $t^+$. One speaks of an inextensible Cosserat surface if the thickness $t = t^+ - t^-$ remains constant for all configurations.

Representations of a configuration in local coordinates are denoted by the same symbol as the configuration itself. We write $\boldsymbol{\xi} = \boldsymbol{\gamma} + \zeta \boldsymbol{\eta}$ for a representation of an admissible shell configuration, which is defined on some parameter domain

$$\mathcal{D}_{[t^-, t^+]} = \{(\vartheta^1, \vartheta^2, \vartheta^3) \in \mathbb{R}^3 \mid$$
$$(\vartheta^1, \vartheta^2) \in \mathcal{D}, \, t^-(\vartheta^1, \vartheta^2) \leq \vartheta^3 \leq t^+(\vartheta^1, \vartheta^2)\}.$$

Here, $\mathcal{D} \subset \mathbb{R}^2$ is an open subset in two-dimensional coordinate space corresponding to a part of the mid-surface. Note that we have $t^- = -t^+$ and $t = 2t^+$ since the reference surface is chosen to be the mid-surface. The thickness is said to be homogeneous if $t$ is constant over the whole surface.

### 2.1.2 Equilibrium equations for thin elastic shells

The governing equations for an elastic shell in static equilibrium follow from the principle of virtual work. The total work done on the system is given by the potential energy

$$V(\boldsymbol{\gamma}) = \int_{\mathcal{A}} W(\boldsymbol{E}(\boldsymbol{\gamma})) \, \mathrm{dA} - \int_{\mathcal{A}} \boldsymbol{F}_{\text{ext}} \cdot \boldsymbol{\gamma} \, \mathrm{dA} - \int_{\partial \mathcal{A}} \boldsymbol{G}_{\text{ext}} \cdot \boldsymbol{\gamma} \, \mathrm{dS},$$

where $\boldsymbol{F}_{\text{ext}}$ is an external force acting on the mid-surface $\mathcal{A}$, $\boldsymbol{G}_{\text{ext}}$ is an external force acting on the boundary $\partial \mathcal{A}$, and $W$ is the stored energy density function, which depends on the strain tensor $\boldsymbol{E}(\boldsymbol{\gamma})$ corresponding to the mid-surface configuration. A shell of Koiter's type has a stored energy density function that consists of a membrane and a bending part:

$$W = \frac{1}{2} \left( \boldsymbol{S}_{\mathrm{m}} : \boldsymbol{E}_{\mathrm{m}} + \boldsymbol{S}_{\mathrm{b}} : \boldsymbol{E}_{\mathrm{b}} \right). \tag{2}$$

It can be derived from three-dimensional elasticity by expressing the strain tensor in terms of kinematic variables of the mid-surface and integrating through the thickness of the shell, assuming a sufficiently thin shell and a small mid-surface strain (Koiter, 1966; Bischoff et al., 2004; Ciarlet, 2005; Steigmann, 2013). The membrane and bending strains are obtained by considering the expansion

$$\boldsymbol{E} = \boldsymbol{E}_{\mathrm{m}} + \zeta \boldsymbol{E}_{\mathrm{b}} + \mathcal{O}(\zeta^2) \tag{3}$$





and taking the in-plane components, while the effective stress resultants are given by

$$\boldsymbol{S}_{\mathrm{m}} = t\boldsymbol{K} : \boldsymbol{E}_{\mathrm{m}}, \quad \boldsymbol{S}_{\mathrm{b}} = \frac{t^3}{12}\boldsymbol{K} : \boldsymbol{E}_{\mathrm{b}}, \tag{4}$$

called the membrane force and the bending moment, respectively. We assume a St.-Venant–Kirchhoff model for linear elastic
isotropic materials, so that the elasticity tensor reads

$$\boldsymbol{K} = \frac{E}{(1-\nu^2)} \begin{bmatrix} 1 & \nu & 0 \\ \nu & 1 & 0 \\ 0 & 0 & (1-\nu)/2 \end{bmatrix} \tag{5}$$

in Voigt notation, assuming the vanishing transverse normal stress condition with Young's modulus $E$ and Poisson's ratio $\nu$.

In a state of equilibrium, the virtual work vanishes, and for any virtual displacement $\delta\boldsymbol{\gamma}$ consistent with the constraints
imposed on the shell, we have that $\delta V = 0$ or, equivalently,

$$\int_{\mathcal{A}} (\boldsymbol{S}_{\mathrm{m}} : \delta\boldsymbol{E}_{\mathrm{m}} + \boldsymbol{S}_{\mathrm{b}} : \delta\boldsymbol{E}_{\mathrm{b}})\, \mathrm{dA} = \int_{\mathcal{A}} \boldsymbol{F}_{\mathrm{ext}} \cdot \delta\boldsymbol{\gamma}\, \mathrm{dA} + \int_{\partial\mathcal{A}} \boldsymbol{G}_{\mathrm{ext}} \cdot \delta\boldsymbol{\gamma}\, \mathrm{dS}.$$

The resulting equation is referred to as the weak variational formulation for a Koiter shell in static equilibrium. It is the starting
point for the numerical solution of variational problems using finite element methods.

We consider the displacement field $\boldsymbol{u} = \boldsymbol{\gamma} - \boldsymbol{\gamma}_0$ of the mid-surface corresponding to an initial undeformed configuration $\boldsymbol{\gamma}_0$
with $\boldsymbol{E}_{\mathrm{m}}(\boldsymbol{\gamma}_0) = \boldsymbol{E}_{\mathrm{b}}(\boldsymbol{\gamma}_0) = 0$ and replace the strain tensors and corresponding effective stress resultants with

$$\boldsymbol{e}_{\mathrm{m}}(\boldsymbol{u}) = \delta\boldsymbol{E}_{\mathrm{m}}(\boldsymbol{\gamma}_0, \boldsymbol{u}), \quad \boldsymbol{e}_{\mathrm{b}}(\boldsymbol{u}) = \delta\boldsymbol{E}_{\mathrm{b}}(\boldsymbol{\gamma}_0, \boldsymbol{u}),$$

$$\boldsymbol{s}_{\mathrm{m}}(\boldsymbol{u}) = t\boldsymbol{K} : \boldsymbol{e}_{\mathrm{m}}(\boldsymbol{u}), \quad \boldsymbol{s}_{\mathrm{b}}(\boldsymbol{u}) = \frac{t^3}{12}\boldsymbol{K} : \boldsymbol{e}_{\mathrm{b}}(\boldsymbol{u}), \tag{6}$$

to obtain the linearized Koiter shell equations. The linearized strain tensors in local curvilinear coordinates read

$$[\boldsymbol{e}_{\mathrm{m}}(\boldsymbol{u})]_{\alpha\beta} = \frac{1}{2}(\partial_\alpha\boldsymbol{u} \cdot \partial_\beta\boldsymbol{\gamma}_0 + \partial_\alpha\boldsymbol{\gamma}_0 \cdot \partial_\beta\boldsymbol{u}),$$

$$[\boldsymbol{e}_{\mathrm{b}}(\boldsymbol{u})]_{\alpha\beta} = -\partial_\alpha\partial_\beta\boldsymbol{u} \cdot \boldsymbol{n}_0 - \partial_\alpha\partial_\beta\boldsymbol{\gamma}_0 \cdot \delta\boldsymbol{n}, \tag{7}$$

for $\alpha, \beta \in \{1,2\}$ and $\delta\boldsymbol{n} = \boldsymbol{m} - (\boldsymbol{n}_0 \cdot \boldsymbol{m})\boldsymbol{n}_0$ with

$$\boldsymbol{n}_0 = \frac{\partial_1\boldsymbol{\gamma}_0 \times \partial_2\boldsymbol{\gamma}_0}{\|\partial_1\boldsymbol{\gamma}_0 \times \partial_2\boldsymbol{\gamma}_0\|}, \quad \boldsymbol{m} = \frac{\partial_1\boldsymbol{u} \times \partial_2\boldsymbol{\gamma}_0 + \partial_1\boldsymbol{\gamma}_0 \times \partial_2\boldsymbol{u}}{\|\partial_1\boldsymbol{\gamma}_0 \times \partial_2\boldsymbol{\gamma}_0\|}.$$

To simplify and fit the problem into an abstract variational framework, we introduce the following notation:

$$a(\boldsymbol{u}, \boldsymbol{v}) = \int_{\mathcal{A}} (\boldsymbol{s}_{\mathrm{m}}(\boldsymbol{u}) : \boldsymbol{e}_{\mathrm{m}}(\boldsymbol{v}) + \boldsymbol{s}_{\mathrm{b}}(\boldsymbol{u}) : \boldsymbol{e}_{\mathrm{b}}(\boldsymbol{v}))\, \mathrm{dA},$$

$$\ell(\boldsymbol{v}) = \int_{\mathcal{A}} \boldsymbol{F}_{\mathrm{ext}} \cdot \boldsymbol{v}\, \mathrm{dA} + \int_{\partial\mathcal{A}} \boldsymbol{G}_{\mathrm{ext}} \cdot \boldsymbol{v}\, \mathrm{dS}. \tag{8}$$

The elastostatic boundary value problem then reads: Find an admissible displacement $\boldsymbol{u}$ such that the equation $a(\boldsymbol{u}, \boldsymbol{v}) = \ell(\boldsymbol{v})$
holds for all admissible variations $\boldsymbol{v} = \delta\boldsymbol{u}$.



### 2.1.3 Reduction to plates and beams

For a Kirchhoff–Love plate, the undeformed configuration is given by a flat surface, e.g., in the $xy$-plane, and only vertical displacements are allowed, i.e.,

$$
\begin{aligned}
\boldsymbol{\gamma}_0(x,y) &= (x,y,0), \\
\boldsymbol{\gamma}(x,y) &= (x,y,w(x,y)), \\
\boldsymbol{u}(x,y) &= (0,0,w(x,y)).
\end{aligned}
\tag{9}
$$

In this case, the membrane part of the strain tensor vanishes and the bending term can be written in terms of the deflection $w$ perpendicular to the plane:

$$
\boldsymbol{s}_{\mathrm{b}}(w) : \boldsymbol{e}_{\mathrm{b}}(v) = D\left(\nu \Delta w \Delta v + (1-\nu)\nabla^2 w : \nabla^2 v\right),
\tag{10}
$$

where $v = \delta w$ is now the variation in vertical direction and the coefficient

$$
D = \frac{Et^3}{12(1-\nu^2)}
\tag{11}
$$

denotes the flexural rigidity of the plate. The weak formulation is then modified to

$$
\begin{aligned}
a(w,v) &= \int_{\mathcal{A}} D\left(\nu \Delta w \Delta v + (1-\nu)\nabla^2 w : \nabla^2 v\right) \mathrm{dA}, \\
\ell(v) &= \int_{\mathcal{A}} f_{\mathrm{ext}}\, v\, \mathrm{dA} + \int_{\partial \mathcal{A}} g_{\mathrm{ext}}\, v\, \mathrm{dS},
\end{aligned}
\tag{12}
$$

where $f_{\mathrm{ext}}$ and $g_{\mathrm{ext}}$ denote the vertical components of $F_{\mathrm{ext}}$ and $G_{\mathrm{ext}}$, respectively.

In the one-dimensional case with $w = w(x)$, the bending term reduces further, which leads to a fourth-order differential equation for an Euler–Bernoulli beam when considering the strong formulation of the problem without boundary conditions:

$$
\frac{\mathrm{d}^2}{\mathrm{d}x^2}\left(\widetilde{D}\,\frac{\mathrm{d}^2 w}{\mathrm{d}x^2}\right) = f_{\mathrm{ext}}.
\tag{13}
$$

Here, $\widetilde{D}$ is another flexural rigidity that depends on the geometry of the beam.

## 2.2 Topographic loading and buoyancy

We model the lithosphere as a thin elastic plate of effective elastic thickness $t$ floating on the asthenosphere and subject to gravitational forces. The initial depth of the mid-surface in the undeformed configuration corresponds to the theoretical depth relative to the mean sea level when there is no overlying mass. The actual mid-surface of the lithosphere does not have to coincide with the Mohorovičić surface, which is the boundary between the crust and the upper mantle of the Earth. However,

we assume that they are close to each other and differ only by a constant vertical displacement.

Starting from the equilibrium equations for a Kirchhoff–Love plate with external load $f_{\mathrm{ext}}$, we split up the contributions from gravity and buoyancy: $f_{\mathrm{ext}} = f_{\mathrm{grav}} + f_{\mathrm{buoy}}$.



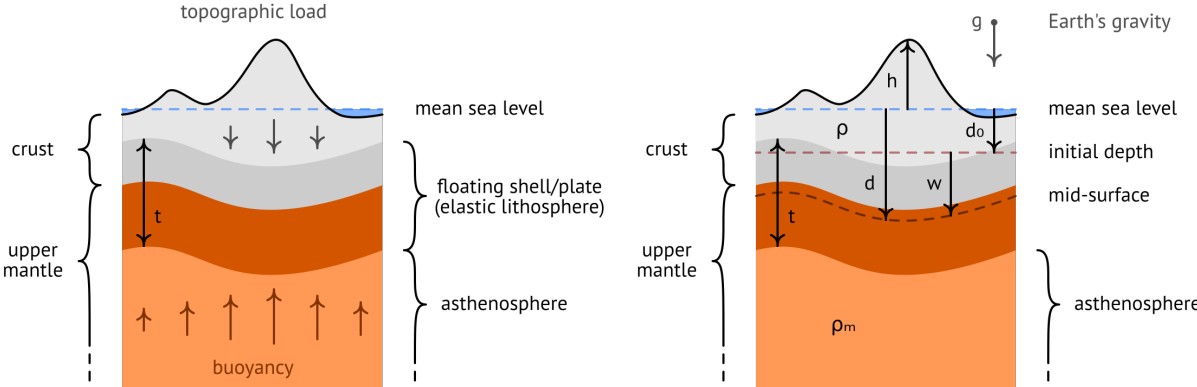

**Figure 4.** Floating elastic lithosphere under topographic loading (left) and relevant quantities for the mathematical model (right).

Gravitational load is obtained by integrating all the weight above the mid-surface. The density of overlying air is considered to be negligible, so that the weight of topography ranges from Earth's surface down to the mid-surface. It is given by

$$f_{\text{grav}} = -\int_d^h \varrho g \, \mathrm{d}z, \tag{14}$$

where $d$ is the depth of the mid-surface relative to the mean sea level, $h$ is the topographic elevation, $\varrho$ is the density of overlying mass, and $g$ is the gravitational acceleration, which is assumed to be constant for the sake of simplicity.

The vertical displacement of the mid-surface from the initial depth is given by $w = d - d_0$. The buoyant force is equal to the weight of displaced asthenosphere, thus

$$f_{\text{buoy}} = \int_d^{d_0} \varrho_{\text{m}} g \, \mathrm{d}z = -\varrho_{\text{m}} g w, \tag{15}$$

assuming a constant upper mantle density $\varrho_{\text{m}}$.

Instead of working with the actual topographic elevation, we use a mass representation $r$ obtained by taking the mass above the mid-surface of the lithosphere and normalizing it by some depth-independent reference density $\varrho_{\text{r}}$. We choose the reference density as the mean rock density from the current depth of the mid-surface to its initial depth and assume that it is homogeneous in space, so that

$$\varrho_{\text{r}} = \frac{1}{|w|} \int_d^{d_0} \varrho \, \mathrm{d}z, \quad r = \frac{1}{\varrho_{\text{r}}} \int_{d_0}^h \varrho \, \mathrm{d}z. \tag{16}$$

Using the mass representation, which corresponds to rock-equivalent topography, we can write the external load as

$$f_{\text{ext}} = -(\varrho_{\text{m}} - \varrho_{\text{r}})gw - \varrho_{\text{r}} g r. \tag{17}$$





Plugging this into the weak variational formulation for the linearized Kirchhoff–Love plate without external boundary forces
yields $a(w,v) + b(w,v) = c(v)$ with the bilinear form in Eq. (12) and

$$b(w,v) = \int_{\mathcal{A}} (\varrho_{\mathrm{m}} - \varrho_{\mathrm{r}}) g w v \, \mathrm{dA}, \quad c(v) = - \int_{\mathcal{A}} \varrho_{\mathrm{r}} g r v \, \mathrm{dA}. \tag{18}$$

This is the Vening-Meinesz model of flexural isostasy used to explain regional compensation (Vening-Meinesz, 1931; Abd-Elmotaal, 1995; Pelletier, 2008, Chapter 5).

In the case where there is no flexural rigidity, the Vening-Meinesz model reduces to the Airy–Heiskanen model of local
isostasy (Airy, 1855), for which the well-known relation

$$\left( \frac{\varrho_{\mathrm{m}} - \varrho_{\mathrm{r}}}{\varrho_{\mathrm{r}}} \right) (d_0 - d) = r \tag{19}$$

holds. It states that the lithospheric depression relative to the initial depth is proportional to the mass representation of the topography with a scaling factor of $(\varrho_{\mathrm{m}} - \varrho_{\mathrm{r}})/\varrho_{\mathrm{r}}$. Using the above relation, we can determine the initial depth from some standard crustal thickness $t_0$ corresponding to a lithospheric plate in local isostasy when the topographic elevation is zero.

### 2.3  Isostatic boundary value problem

If we consider only a portion of Earth's lithosphere for the simulation, conditions on the boundary of the domain have to be prescribed to compensate for the missing information outside of it. A natural choice is given by the full Neumann boundary condition, which corresponds to setting $g_{\mathrm{ext}} = 0$ on the whole boundary. The resulting isostatic boundary value problem for a Kirchhoff–Love plate then reads: Find an admissible deflection $w$ such that $a(w,v) + b(w,v) = c(v)$ for all admissible variations
$v = \delta w$.

The Sobolev space $\mathrm{H}^2(\mathcal{A})$ is chosen as the space of admissible deflections for the boundary value problem. It consists of square-integrable functions on the reference surface with square-integrable weak derivatives up to second order. For the full Neumann problem, it can be shown using Korn's inequality that the variational problem is well-posed, provided that $\mathcal{A}$ is a bounded domain with piecewise smooth boundary and the coefficient $(\varrho_{\mathrm{m}} - \varrho_{\mathrm{r}})g$ in the buoyancy term is bounded from below
by a positive number.

The above displacement formulation requires $\mathrm{H}^2$ regularity, which implies global $\mathrm{C}^1$ continuity for the trial and test functions. The difficulty of $\mathrm{C}^1$ finite elements can be circumvented by considering isogeometric shape functions.

### 2.4  Spherical model of the lithosphere

Using the more general shell equations, it is possible to perform simulations of the lithosphere on the whole surface of the Earth.
From a modeling point of view, the results may not reflect the physical reality since the Earth consists of different regimes and tectonic plates that interact with each other in a complex manner. Furthermore, due to the large scale of the simulation, the effects of flexural rigidity will not be visible. Nevertheless, we assume that the entire lithosphere can be modeled as a single spherical shell to showcase the capabilities of isogeometric analysis in numerical simulations on curved domains, especially



on a spherical domain. Note that it is also possible to model the surface of the Earth as an oblate spheroid or an irregular geoid
instead of a sphere using isogeometric analysis. For the sake of simplicity, we restrict ourselves to the spherical model in this
paper.

Some considerations in Sect. 2.2 for lithospheric plates in isostatic equilibrium have to be adapted to the shell model. The
buoyant force in three dimensions reads

$$b(\boldsymbol{u},\boldsymbol{v}) = \int_{\mathcal{A}} (\varrho_{\mathrm{m}} - \varrho_{\mathrm{r}})g(\boldsymbol{n}\cdot\boldsymbol{u})(\boldsymbol{n}\cdot\boldsymbol{v})\,\mathrm{dA}, \tag{20}$$

where $(\boldsymbol{n}\cdot\boldsymbol{u})$ and $(\boldsymbol{n}\cdot\boldsymbol{v})$ are the radial part of the trial and test function, respectively, given by the orthogonal projection onto
the unit normal $\boldsymbol{n}$ of the sphere. Similarly, the external load is given by a radial gravitational force

$$c(\boldsymbol{v}) = -\int_{\mathcal{A}} \varrho_{\mathrm{r}}gr(\boldsymbol{n}\cdot\boldsymbol{v})\,\mathrm{dA}. \tag{21}$$

With the above adjustments, the isostatic problem for a Koiter shell then reads: Find an admissible displacement $\boldsymbol{u}$ such that
$a(\boldsymbol{u},\boldsymbol{v}) + b(\boldsymbol{u},\boldsymbol{v}) = c(\boldsymbol{v})$ holds for all admissible variations $\boldsymbol{v} = \delta\boldsymbol{u}$. We consider the vector-valued Sobolev space $\mathrm{H}^2(\mathcal{A})^3$ for
the displacements of the spherical shell.

## 3  Isogeometric finite element analysis

Isogeometric analysis (IGA) is a computational approach for solving partial differential equations (PDE) numerically that em-
ploys non-uniform rational basis splines (NURBS) to both parametrize the domain and construct finite element approximations
of solutions to the equations. This section introduces the notions required for the numerical discretization of elliptic boundary
value problems using isogeometric analysis, in particular the isostatic boundary value problem. We begin with the definition of
B-splines and NURBS. A more elaborate treatment of NURBS with numerical algorithms can be found in the NURBS book
(Piegl and Tiller, 1995).

### 3.1  B-splines and NURBS

Let $\vartheta_0 \leq \cdots \leq \vartheta_m$ be a finite sequence of non-decreasing real numbers. A spline of degree $p$ is a piecewise polynomial function
$f\colon [\vartheta_0,\vartheta_m] \to \mathbb{R}$ with the property that the restriction to each subinterval $[\vartheta_{i-1},\vartheta_i)$ for $i = 1,\ldots,m$ is a polynomial function
of maximum degree $p$. The tuple $\boldsymbol{\Theta} = (\vartheta_0,\ldots,\vartheta_m)$ is called a knot sequence for the spline with knot values $\vartheta_i$ and the term
breakpoint is used to refer to a distinct knot value. The half-open interval $[\vartheta_{i-1},\vartheta_i)$ is called the $i$-th knot span, which can be
empty.

The maximum order of continuity that a spline of degree $p$ can attain at the breakpoints is $p-1$. We refer to such splines as
smooth splines. A lower order of continuity can be obtained by placing multiple knots in the same location. Each additional
knot reduces the order of continuity by one until the resulting spline is discontinuous at the breakpoint.

The type of a spline is completely characterized by its degree and the knot sequence. Let $\mathcal{S}(\boldsymbol{\Theta},p)$ denote the space of splines
of degree $p$ with knot sequence $\boldsymbol{\Theta}$. It is a vector space of dimension $m-p$. By introducing the numbers $\widetilde{m} = m+1$, $\widetilde{p} = p+1$,





and $\widetilde{n} = n + 1$, where $\widetilde{m}$ is the number of knots, $\widetilde{p}$ is the order of the spline, and $\widetilde{n}$ is the dimension of $\mathcal{S}(\boldsymbol{\Theta}, p)$, we can write

$\widetilde{n} = \widetilde{m} - \widetilde{p}$ or, equivalently, $m = n + p + 1$.

In the following, we consider splines on the unit interval $[0, 1]$ with an open knot sequence, i.e., the first and last knot values have multiplicity $p + 1$. Then the knot sequence has the form

$$\boldsymbol{\Theta} = (\underbrace{0, \ldots, 0}_{(p+1)\text{-times}}, \vartheta_{p+1}, \ldots, \vartheta_n, \underbrace{1, \ldots, 1}_{(p+1)\text{-times}}),$$

where we have $\vartheta_0 = \cdots = \vartheta_p = 0$, $\vartheta_{n+1} = \cdots = \vartheta_{n+p+1} = 1$ and $\vartheta_i \in (0, 1)$ for $i = p+1, \ldots, n$.

### 250 3.1.1 B-spline basis functions

A particular basis for the spline space $\mathcal{S}(\boldsymbol{\Theta}, p)$ is given by the B-splines (basis splines). They have minimal support and allow for quick evaluation of the splines using de Boor's algorithm, which is convenient for isogeometric analysis. The B-splines of degree $p$ are recursively defined via the Cox–de Boor formula

$$B_{k,p}(\vartheta) = \frac{\vartheta - \vartheta_k}{\vartheta_{k+p} - \vartheta_k} B_{k,p-1}(\vartheta) + \frac{\vartheta_{k+1+p} - \vartheta}{\vartheta_{k+1+p} - \vartheta_{k+1}} B_{k+1,p-1}(\vartheta)$$

with $\vartheta \in [0, 1]$, $k = 0, \ldots, n$, and

$$B_{k,0}(\vartheta) = \begin{cases} 1 & \text{for } \vartheta \in [\vartheta_k, \vartheta_{k+1}), \\ 0 & \text{otherwise,} \end{cases}$$

for $k = 0, \ldots, n + p$, where the convention $0/0 = 0$ is used if the knot values in the denominator coincide. We refer to $B_{k,p}$ as the $k$-th B-spline basis function of degree $p$.

Given a finite sequence of control points $\boldsymbol{c}_0, \ldots, \boldsymbol{c}_n \in \mathbb{R}^r$ in the physical space of dimension $r$, we can construct a B-spline

curve of degree $p$ through a linear combination of the form

$$\boldsymbol{\gamma} \colon [0, 1] \to \mathbb{R}^r, \boldsymbol{\gamma}(\vartheta) = \sum_{k=0}^{n} \boldsymbol{c}_k B_{k,p}(\vartheta).$$

B-spline curves are commonly used to represent shapes in computer-aided geometric design (CAGD). The description using control points allows for intuitive local manipulation of free-form shapes. In isogeometric analysis, the control points additionally serve as degrees of freedom for the unknowns in a discretized system of equations.

If the domain of the problem is two- or three-dimensional, B-spline surfaces or volumes are used to describe its geometry. Multivariate spline spaces are constructed via the tensor product of univariate spline spaces. Instead of a single knot sequence and a single spline degree, we have a family of knot sequences $\boldsymbol{\Theta} = (\boldsymbol{\Theta}^{(1)}, \ldots, \boldsymbol{\Theta}^{(d)})$ along with a tuple of spline degrees $\boldsymbol{p} = (p_1, \ldots, p_d)$ corresponding to each parametric dimension. The B-spline basis functions of the spline space $\mathcal{S}_d(\boldsymbol{\Theta}, \boldsymbol{p}) = \mathcal{S}(\boldsymbol{\Theta}^{(1)}, p_1) \otimes \cdots \otimes \mathcal{S}(\boldsymbol{\Theta}^{(d)}, p_d)$ are given by

$B_{\boldsymbol{k}, \boldsymbol{p}}(\boldsymbol{\vartheta}) = B_{k_1, p_1}^{(1)}(\vartheta_1) \cdots B_{k_d, p_d}^{(d)}(\vartheta_d)$





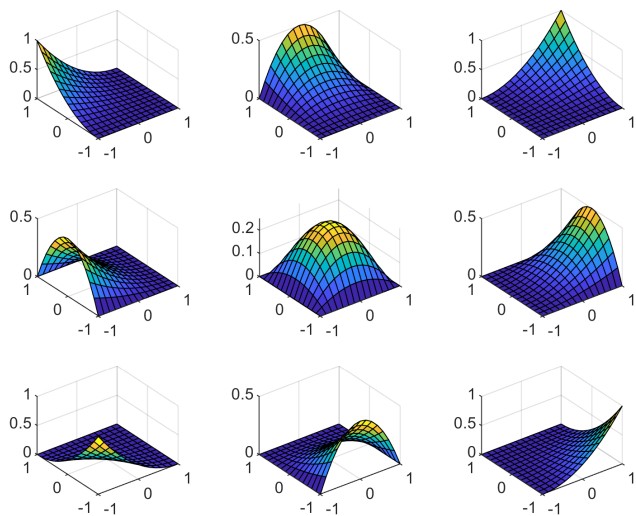

**Figure 5.** Isogeometric shape functions on a bivariate quadratic B-spline patch.

for $\boldsymbol{\vartheta} = (\vartheta_1, \ldots, \vartheta_d) \in [0,1]^d$ in the multivariate setting. The B-splines $B_{0,p_j}^{(j)}, \ldots, B_{n_j,p_j}^{(j)}$ form a basis for $\mathcal{S}(\boldsymbol{\Theta}^{(j)}, p_j)$ and $\boldsymbol{k} = (k_1, \ldots, k_d)$ is a multi-index with $k_j = 0, \ldots, n_j$ for $j = 1, \ldots, d$. We order the basis functions lexicographically, so that $B_{k,\boldsymbol{p}}$ corresponds to the $k$-th basis function when using an integer index $k = 0, \ldots, n$ instead of a multi-index.

A $d$-variate B-spline patch corresponding to the control points $\boldsymbol{c}_0, \ldots, \boldsymbol{c}_n \in \mathbb{R}^r$ is a parametrization of the form

$$\boldsymbol{\gamma} \colon [0,1]^d \to \mathbb{R}^r, \boldsymbol{\gamma}(\boldsymbol{\vartheta}) = \sum_{k=0}^{n} \boldsymbol{c}_k B_{k,\boldsymbol{p}}(\boldsymbol{\vartheta}).$$

We refer to the image of $\boldsymbol{\gamma}$ also as a B-spline patch and write $\mathcal{S}_{d,r}(\boldsymbol{\Theta}, \boldsymbol{p})$ for the space of $d$-variate B-spline patches in $\mathbb{R}^r$.

### 3.1.2 NURBS basis functions

B-splines can be generalized to include rational functions in addition to polynomial ones by assigning a weight to each control point. This greatly increases the design capabilities of free-form shapes, e.g., conic sections can be exactly represented by rational B-splines with weighted control points as opposed to non-rational ones. The term non-uniform in the acronym NURBS stresses the fact that the distribution of knot values in the knot sequence is not necessarily uniform.

Given a B-spline basis $B_{0,p}, \ldots, B_{n,p}$ for the spline space $\mathcal{S}(\boldsymbol{\Theta}, p)$ and a tuple of positive weights $\boldsymbol{\omega} = (\omega_0, \ldots, \omega_n)$, the NURBS space $\mathcal{S}^{\boldsymbol{\omega}}(\boldsymbol{\Theta}, p)$ is generated by rational functions of the form

$$N_{k,p}^{\boldsymbol{\omega}}(\vartheta) = \frac{\omega_k B_{k,p}(\vartheta)}{\omega(\vartheta)}, \quad k = 0, \ldots, n,$$

with $\vartheta \in [0,1]$. The weight function in the denominator is a weighted sum of the B-spline basis functions

$$\omega(\vartheta) = \sum_{k=0}^{n} \omega_k B_{k,p}(\vartheta).$$





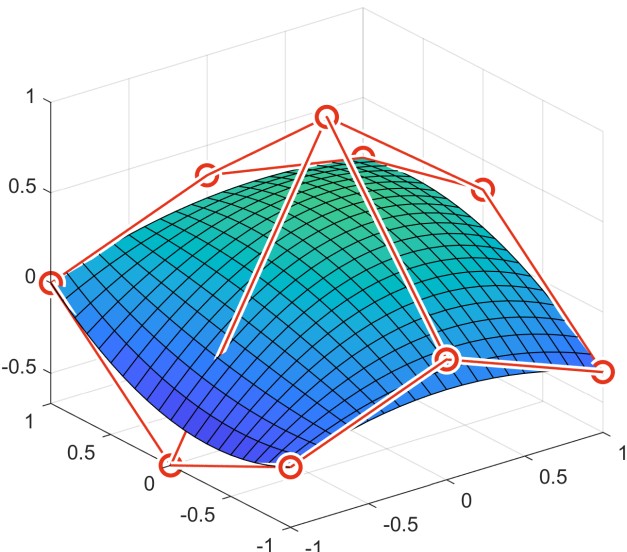

**Figure 6.** Linear combination of the isogeometric shape functions in Fig. 5 and corresponding mesh of control points (red).

Note that the original spline space is a special case of the NURBS space with constant weights. For the multivariate case, we proceed similarly to the non-rational B-splines and define the NURBS basis functions as

$$N_{k,\boldsymbol{p}}^{\boldsymbol{\omega}}(\boldsymbol{\vartheta}) = \frac{\prod_{j=1}^{d} \omega_{k_j}^{(j)} B_{k_j,p_j}^{(j)}(\vartheta_j)}{\omega(\boldsymbol{\vartheta})}, \quad k = 0, \dots, n,$$

for $\boldsymbol{\vartheta} = (\vartheta_1, \dots, \vartheta_d) \in [0,1]^d$. The multivariate weight function reads

$$\omega(\boldsymbol{\vartheta}) = \sum_{k_1=0}^{n_1} \cdots \sum_{k_d=0}^{n_d} \omega_{k_1}^{(1)} B_{k_1,p_1}^{(1)}(\vartheta_1) \cdots \omega_{k_d}^{(d)} B_{k_d,p_d}^{(d)}(\vartheta_d),$$

where $\boldsymbol{\omega} = (\boldsymbol{\omega}^{(1)}, \dots, \boldsymbol{\omega}^{(d)})$ is a family of weight tuples corresponding to each parametric dimension. Contrary to non-rational B-splines, the resulting multivariate NURBS space $\mathcal{S}_d^{\boldsymbol{\omega}}(\boldsymbol{\Theta}, \boldsymbol{p})$ is no longer a tensor product space because of the weight function $\omega$. Nevertheless, it is called tensor-product-like, following the terminology in isogeometric analysis.

A $d$-variate NURBS patch corresponding to the control points $\boldsymbol{c}_0, \dots, \boldsymbol{c}_n \in \mathbb{R}^r$ is a parametrization of the form

$$\boldsymbol{\gamma} \colon [0,1]^d \to \mathbb{R}^r, \boldsymbol{\gamma}(\boldsymbol{\vartheta}) = \sum_{k=0}^{n} \boldsymbol{c}_k N_{k,\boldsymbol{p}}^{\boldsymbol{\omega}}(\boldsymbol{\vartheta}).$$

We refer to the image of $\boldsymbol{\gamma}$ also as a NURBS patch and write $\mathcal{S}_{d,r}^{\boldsymbol{\omega}}(\boldsymbol{\Theta}, \boldsymbol{p})$ for the space of $d$-variate NURBS patches in $\mathbb{R}^r$.

To shorten the notation, we omit the superscript $\boldsymbol{\omega}$ and the subscript $\boldsymbol{p}$ from the NURBS basis functions and introduce the double index $\alpha = (k,l)$, ranging from $(0,1)$ to $(n,r)$, so that a NURBS patch can be written as

$$\boldsymbol{\gamma} = \sum_{k=0}^{n} \sum_{l=1}^{r} c_{k,l} N_k \boldsymbol{e}_l = \sum_{k=0}^{n} \sum_{l=1}^{r} c_{k,l} \boldsymbol{N}_{k,l} = \sum_{\alpha \in \mathfrak{A}} c_\alpha \boldsymbol{N}_\alpha,$$



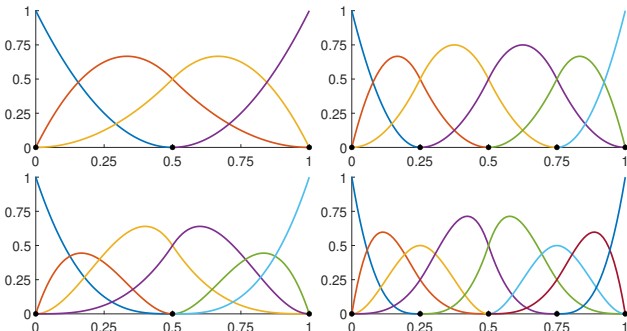

**Figure 7.** Initial isogeometric shape functions (quadratic B-splines, top-left) and the shape functions that result from $h$-refinement (top-right), $p$-refinement (bottom-left), and $k$-refinement (bottom-right).

where $\mathfrak{A} = \left\{ (k,l) \in \mathbb{N}_0^2 \mid 0 \leq k \leq n, 1 \leq l \leq r \right\}$ is the index set with $(n+1) \cdot r$ elements, $c_\alpha = c_{k,l}$ is the $l$-th component of the $k$-th control point, and $\boldsymbol{N}_\alpha = \boldsymbol{N}_{k,l} = N_k \boldsymbol{e}_l$ is the $\alpha$-th vector-valued NURBS basis function.

### 3.1.3 Refinement methods

In order to get better approximation results for the numerical solutions, the NURBS space used for the discretization of the
problem has to be refined. There are two refinement methods that increase the number of shape functions and maintain the global smoothness of the NURBS space. The first method is called knot insertion, also known as $h$-refinement, where a finer NURBS space is constructed by adding new breakpoints to the knot sequence. The second one is order elevation, or $p$-refinement, which raises the order of the NURBS space without changing the knot spans. Performing order elevation followed by knot insertion results in a so-called $k$-refinement. See Fig. 7 for an illustration of the methods.

For the multivariate case, inserting a breakpoint to a knot sequence will affect all elements along the transverse direction due to the tensor-product-like structure. To enable local refinement, several methods can be considered. In our work, we employ hierarchical B-splines as described in Vuong et al. (2011). Adaptive local refinement can then be performed if an error estimator for the numerical solution to the problem is available (Garau and Vázquez, 2018; Buffa et al., 2022).

### 3.2 Isogeometric discretization

Given a weak formulation of the variational problem, a numerical solution can be obtained by considering a projection onto some finite-dimensional subspace of the solution space. This is generally referred to as a Galerkin projection. Finite element methods are based on subdividing the domain of the problem into finitely many elements, on which a number of shape functions are defined. The finite element space is then constructed from linear combinations of the shape functions on each element that satisfy certain interpolation conditions.

Isoparametric finite elements enable the solution of problems on domains with curved boundaries by using the same shape functions for the numerical approximation of solutions to describe the geometry of the domain. They serve as a basis for the



isogeometric paradigm, where we consider domains that can be represented by some NURBS geometry and use refinements of the corresponding NURBS space to construct approximations of solutions to the problem.

An isogeometric mesh consists of NURBS patches, each of which can be refined to increase the accuracy of the numerical approximation. As opposed to a standard finite element mesh, the smoothness of shape functions within each patch can be preserved without much effort when the mesh is subdivided into smaller elements. This greatly reduces the number of degrees of freedom compared to classical $C^1$ finite elements, which is useful when working with shell and plate equations that require global $C^1$ continuity.

We first consider domains that can be exactly represented by a single NURBS patch. The main idea is to transform the 330 problem posed on the patch to a fixed parameter domain, approximate the solution with a linear combination of NURBS functions that result from refinements of the NURBS space associated with the geometry function, and transform the numerical solution back to the physical domain.

### 3.2.1 Domain transformation

Let $\Omega \subset \mathbb{R}^r$ denote the physical domain described by the geometry function

$$\boldsymbol{\gamma}_0 \colon \hat{\Omega} \to \Omega, \, \boldsymbol{\gamma}_0(\boldsymbol{\vartheta}) = \sum_{k=0}^n \sum_{l=1}^r G_{k,l} \boldsymbol{N}_{k,l}(\boldsymbol{\vartheta}), \tag{22}$$

with control points $\boldsymbol{G}_0, \dots, \boldsymbol{G}_n \in \mathbb{R}^r$ and the parameter domain $\hat{\Omega} = [0,1]^d$. To ensure that the domain is suitable for isogeometric analysis, we require that the geometry function is at least a bi-Lipschitz transformation. Thus, it is important to impose conditions on the control points of the geometry such that this requirement is fulfilled.

The weak formulation of a variational problem posed on the physical domain can be transformed to the parameter domain 340 by pulling back functions in the solution space $\mathcal{V}(\Omega)$ to the parameter domain using the geometry function. By doing so, we obtain an equivalent weak formulation of the problem on the parameter domain: Find $\hat{\boldsymbol{u}} \in \hat{\mathcal{V}}(\hat{\Omega})$ such that $\hat{a}(\hat{\boldsymbol{u}}, \hat{\boldsymbol{v}}) = \hat{\ell}(\hat{\boldsymbol{v}})$ for all $\hat{\boldsymbol{v}} \in \hat{\mathcal{V}}(\hat{\Omega})$ with $\hat{\mathcal{V}}(\hat{\Omega}) = \{\boldsymbol{v} \circ \boldsymbol{\gamma}_0 \mid \boldsymbol{v} \in \mathcal{V}(\Omega)\}$ and

$$\hat{a}(\hat{\boldsymbol{u}}, \hat{\boldsymbol{v}}) = a(\hat{\boldsymbol{u}} \circ \boldsymbol{\gamma}_0^{-1}, \hat{\boldsymbol{v}} \circ \boldsymbol{\gamma}_0^{-1}), \quad \hat{\ell}(\hat{\boldsymbol{v}}) = \ell(\hat{\boldsymbol{v}} \circ \boldsymbol{\gamma}_0^{-1}).$$

### 3.2.2 Ritz–Galerkin method

To discretize the transformed weak formulation, we consider isogeometric shape functions that result from refinements of the NURBS space associated with the geometry function. We choose $\hat{\mathcal{V}}_{h,p} = \mathcal{S}_{d,s}^{\boldsymbol{\omega}}(\boldsymbol{\Theta}_h, \boldsymbol{p}) \cap \hat{\mathcal{V}}(\hat{\Omega})$ for the finite-dimensional subspace, where $h$ is a discretization parameter corresponding to the diameter of elements and $s$ is the number of components of functions in the solution space. Galerkin projection then yields a family of finite-dimensional problems of the form: Find $\hat{\boldsymbol{u}}_{h,p} \in \hat{\mathcal{V}}_{h,p}$ such that $\hat{a}(\hat{\boldsymbol{u}}_{h,p}, \hat{\boldsymbol{v}}_{h,p}) = \hat{\ell}(\hat{\boldsymbol{v}}_{h,p})$ for all $\hat{\boldsymbol{v}}_{h,p} \in \hat{\mathcal{V}}_{h,p}$.

The trial and test functions are now given by linear combinations of the NURBS basis functions, i.e.,

$$\hat{\boldsymbol{u}}_{h,p} = \sum_{k=0}^n \sum_{l=1}^s U_{k,l} \boldsymbol{N}_{k,l}, \quad \hat{\boldsymbol{v}}_{h,p} = \sum_{k=0}^n \sum_{l=1}^s V_{k,l} \boldsymbol{N}_{k,l}, \tag{23}$$



with control points $\boldsymbol{U}_0, \ldots, \boldsymbol{U}_n \in \mathbb{R}^s$ and $\boldsymbol{V}_0, \ldots, \boldsymbol{V}_n \in \mathbb{R}^s$, respectively. We can stack the control points on top of each other and turn each of the two sets of control points into a single column vector so that the Galerkin equation can be written in matrix-vector form. The problem then reduces to solving a system of linear equations of the form $\boldsymbol{A}^\top \boldsymbol{U} = \boldsymbol{L}$ with the coefficient matrix $A_{\alpha\beta} = \hat{a}(\boldsymbol{N}_\alpha, \boldsymbol{N}_\beta)$ and the right-hand side $L_\beta = \hat{\ell}(\boldsymbol{N}_\beta)$, where $\alpha$ and $\beta$ are double indices in some specified order, ranging from $(0,1)$ to $(n,s)$.

The solution vector $\boldsymbol{U} \in \mathbb{R}^{(n+1)s}$ contains the coordinates of the control points associated with the trial function $\hat{\boldsymbol{u}}_{h,p}$ and is referred to as the vector of degrees of freedom in the fully unconstrained solution space. When boundary or interface conditions are present, it is restricted to a subspace fulfilling those conditions.

### 3.2.3   Multi-patch $\mathrm{C}^1$ coupling

In the case where the domain consists of multiple NURBS patches, isogeometric shape functions at the patch interfaces have to be adjusted to maintain the global $\mathrm{C}^1$ smoothness. There are various methods that can be employed to achieve this. Multi-patch $\mathrm{C}^1$ isogeometric spline spaces can be constructed by replacing shape functions at the boundary of each patch that coincides with the boundary of another patch with interface functions that span over multiple patches (Farahat et al., 2023). Another approach is to stitch shape functions at the patch interfaces together by imposing the $\mathrm{C}^1$ condition at some collocation points or weakly via the constraint matrix

$$C_{\alpha\beta} = \int\limits_\Gamma [\![\boldsymbol{n} \cdot \boldsymbol{N}_\alpha]\!][\![\boldsymbol{n} \cdot \boldsymbol{N}_\beta]\!]\,\mathrm{dS}, \tag{24}$$

where $\boldsymbol{n}$ is a unit normal at the patch interface $\Gamma$ and $[\![\cdot]\!]$ denotes the jump of a function between the patches. The latter is a method proposed by Collin et al. (2016) and will be used for the numerical experiments in this work. It has the drawback that the resulting system of linear equations will lose its sparse structure. Aside from that, the computation of the null space corresponding to the $\mathrm{C}^1$ constraint is generally a difficult task numerically.

The construction of multi-patch $\mathrm{C}^1$ isogeometric spline spaces with optimal approximation properties is a challenging problem for complex geometries. A so-called $\mathrm{C}^1$ locking might occur for $\mathrm{G}^1$ multi-patch parametrizations that are not analysis-suitable (Collin et al., 2016). In this work, we will mainly consider planar domains that result from joining convex quadrilaterals along the sides. It has been shown that the class of bilinear $\mathrm{G}^1$ parametrizations is analysis-suitable, so that optimal convergence can be achieved in this setting.

## 4   Parameter identification from measured data

In this section, we describe a method to identify parameters of the plate model that are most plausible to explain the measured data for the Mohorovičić depth. The quantities we are interested in are the effective elastic thickness, the reference density, and the topographic load that acts on the lithosphere. To determine the spatial distribution of those quantities, we perform PDE constrained optimization with a tracking-type objective function, e.g., a quadratic loss function.





## 4.1 Tracking-type optimization problem

A general PDE constrained optimization problem reads

$$\begin{aligned}
\min \quad & J(q,w), \\
\text{s.t.} \quad & R(q,w) = 0,
\end{aligned} \tag{25}$$

where $J$ is the objective function and $R$ is the state equation operator that determines the governing equations. The input consists of the design variable $q$, which represents the sought parameters that are to be optimized, and the state variable $w$, which is a candidate solution to the state equation associated with the design variable.

A tracking-type objective function that is commonly used is given by the integrated squared error

$$J(q,w) = \frac{1}{2} \int_{\mathcal{A}} (w - w_{\mathrm{d}})^2 \, \mathrm{dA}, \tag{26}$$

which corresponds to the method of least squares and gauges the deviation of $w$ from the observed data $w_{\mathrm{d}}$. The state equation of the isostatic problem in weak formulation reads

$$\langle R(q,w), z \rangle = a(q,w,z) + b(p,w,z) - c(r,z) = 0 \tag{27}$$

for all variations $z$, where $\langle \, \cdot \, , \, \cdot \, \rangle$ is the duality pairing and

$$\begin{aligned}
a(q,w,z) &= \int_{\mathcal{A}} q(\nu \Delta w \Delta z + (1 - \nu) \nabla^2 w : \nabla^2 z) \, \mathrm{dA}, \\
b(p,w,z) &= \int_{\mathcal{A}} pwz \, \mathrm{dA}, \quad c(r,z) = - \int_{\mathcal{A}} rz \, \mathrm{dA},
\end{aligned} \tag{28}$$

with the flexural parameter, the crustal depth-to-height ratio, and the rock-equivalent topography given by

$$q = \frac{D}{\varrho_{\mathrm{r}} g} = \frac{Et^3}{12(1 - \nu^2)\varrho_{\mathrm{r}} g}, \quad p = \frac{\varrho_{\mathrm{m}} - \varrho_{\mathrm{r}}}{\varrho_{\mathrm{r}}}, \quad r = \frac{1}{\varrho_{\mathrm{r}}} \int_{d_0}^{h} \varrho \, \mathrm{dz},$$

respectively.

## 4.2 Adjoint state method

Starting from an initial guess for the design variable, the idea is to move in a direction along which the objective function
decreases. Such a direction can be found via the gradient of the reduced cost functional

$$I(q) = J(q, w(q)), \tag{29}$$

which accounts for the dependence of the state variable $w$ on the design variable $q$. An efficient way to evaluate the gradient of $I$ without computing sensitivities of $w$ with respect to $q$ is given by the adjoint state method (Hinze et al., 2008).





The adjoint state equation for the isostatic boundary value problem with integrated squared error as objective function and flexural parameter as design variable reads

$$a(q, \delta w, z) + b(p, \delta w, z) = \int_{\mathcal{A}} (w(q) - w_\mathrm{d}) \delta w \, \mathrm{dA} \tag{30}$$

for all variations $\delta w$. Given a solution $z(q, w(q))$ to the adjoint state equation, which is referred to as an adjoint state of the problem, the first variation of the reduced cost functional in the direction of $\delta q$ can be computed via

$$\delta I(q, \delta q) = -a(\delta q, w(q), z(q, w(q))). \tag{31}$$

The $\mathrm{L}^2$ gradient of $I$ is then characterized by the scalar field $\nabla I(q)$ in the Lebesgue space $\mathrm{L}^2(\mathcal{A})$ that satisfies

$$\int_{\mathcal{A}} \nabla I(q) \delta q \, \mathrm{dA} = \delta I(q, \delta q) \tag{32}$$

for all variations $\delta q$. Similar considerations can be done for the reference density and the rock-equivalent topography as design variables.

### 4.3 Isogeometric optimization

To compute the $\mathrm{L}^2$ gradient numerically, we discretize the variables using isogeometric shape functions and solve for the coefficients of the linear combinations that approximate the sought quantities (see Sect. 3.2.2).

We perform a steepest descent method to find the optimal parameters iteratively. Let $q_k$, $w_k$, and $z_k$ be the discretized variables in the $k$-th step of the optimization procedure. An optimization loop consists of the following steps:

1. Solve the state equation (Eq. 27) for $w_k(q_k)$,

2. Solve the adjoint equation (Eq. 30) for $z_k(q_k, w(q_k))$,

3. Compute the $\mathrm{L}^2$ gradient $\nabla I(q_k)$ via Eqs. (31) and (32),

4. Update the design variable $q_{k+1}$ using $\nabla I(q_k)$.

For the design updates, we apply a backtracking line search based on the Armijo–Goldstein condition along the negative of the gradient

$$q_{k+1} = q_k - s_k \nabla I(q_k), \tag{33}$$

where $s_k$ is the Armijo step size (Hinze et al., 2008).

## 5 Numerical results and discussion

The isostatic boundary value problem for a Kirchhoff–Love plate (see Sect. 2.3) is solved numerically using methods of isogeometric analysis (see Sect. 3). Spectral methods (Nunn and Aires, 1988), finite difference methods (Wickert, 2016), as





| Location | Longitude | Latitude |
|----------|-----------|----------|
| Central Java | 109.5° to 111.75° | -8.5° to -6.25° |
| Java Island | 105° to 115° | -10° to -5° |
| Indonesia | 90° to 150° | -15° to 15° |
| Hawaii | -165° to -150° | 13° to 28° |
| Himalaya | 60° to 120° | 20° to 50° |
| Europe | -25° to 25° | 28° to 78° |

**Table 1.** Geographic coordinates of locations of interest.

well as standard finite element methods (Manríquez et al., 2014) have been commonly used to simulate the lithospheric flexure. The advantage of using isogeometric finite elements lies in the simple construction of smooth shape functions.

We demonstrate our approach on the following locations: Central Java, the Java Island, the Indonesian Archipelago, the Hawaiian Islands, the Himalayan Mountain Range, and the European Plate. The corresponding geographic coordinates in decimal degrees are listed in Table 1.

The Earth2014 data (Hirt and Rexer, 2015) contain rock-equivalent topography that can be converted into topographic load by using the reference density and the gravitational acceleration in Table 2. For Central Java, we consider both a single-patch and multi-patch parametrization of the domain to show the capabilities of multi-patch isogeometric analysis. The results can be compared with the Mohorovičić depth data obtained using inversion of receiver functions from the work of Amukti et al. (2019). A Mohorovičić depth map is available for the European Plate (Grad et al., 2009), which is also used to estimate model

parameters in Sect. 5.2.

### 5.1    Simulation of the lithospheric depression

In this subsection, we compare the results obtained from the simple Airy–Heiskanen model of local isostasy with the regional model of flexural isostasy by Vening-Meinesz to simulate the lithospheric depression due to topographic loading and buoyancy. Isogeometric analysis is used to solve the isostatic boundary value problem for the flexural model numerically. We choose the

physical parameters in Table 2, which are assumed to be constant over the simulation domain. The mesh is subdivided into $16 \times 16$ elements and a spline degree of $(4,4)$ is chosen for the isogeometric spline space.

Figure 8a (left) shows a contour plot of the bedrock topography of Central Java. The corresponding topographic load, expressed through rock-equivalent topography, is shown in Fig. 8a (right), which also contains a multi-patch geometry of the domain of interest.

The computed lithospheric depression for a single-patch domain is shown in Fig. 8b (right). Compared to the Airy–Heiskanen model in Fig. 8b (left) and the available depth data (Amukti et al., 2019, Fig. 6), the topographic loading in the Vening-Meinesz model is additionally compensated by flexural rigidity. This leads to less local variations. High-frequency





| Parameter | Value |
|---|---|
| Young's modulus $E$ | $65\,\mathrm{GPa}$ |
| Poisson's ratio $\nu$ | $0.25$ |
| reference rock density $\varrho_\mathrm{r}$ | $2.67\,\mathrm{g\,cm}^{-3}$ |
| upper mantle density $\varrho_\mathrm{m}$ | $3.33\,\mathrm{g\,cm}^{-3}$ |
| gravitational acceleration $g$ | $9.81\,\mathrm{m\,s}^{-2}$ |
| effective elastic thickness $t$ | $16\,\mathrm{km}$ |
| standard crustal thickness $t_0$ | $30\,\mathrm{km}$ |
| Earth radius $R_\mathrm{E}$ | $6371\,\mathrm{km}$ |

**Table 2.** Physical parameters for the numerical simulations.

details are strongly attenuated and the mid-surface only reaches a depth of less than $32\,\mathrm{km}$ as opposed to the Airy–Heiskanen model that predicts Mohorovičić depths up to $42\,\mathrm{km}$ below the mean sea level when using the same physical parameters.

The result of the multi-patch simulation is depicted in Fig. 8c (left). It differs from the single-patch result due to the missing data outside of the simulation domain that are replaced by Neumann boundary conditions. Augmenting the multi-patch domain with additional patches that cover the whole rectangular single-patch domain yields a result that is close to the single-patch solution (see Fig. 8c, right). Both solutions also appear continuously differentiable at the interfaces and require less computational effort and degrees of freedom than a classical approach using conforming $\mathrm{C}^1$ finite elements or non-conforming discrete

Kirchhoff elements, provided that multiple patches are used sparingly.

Numerical experiments for the other geographic locations have been done to observe the effect of different scales and varying load distributions (see Figs. 9a to 9d). Large scale simulations require more degrees of freedom to resolve tiny details of the solution. Uniform refinement of the mesh leads to a rapid increase in computational effort, which may not be necessary for regions that are already resolved to a sufficient accuracy. In order to reduce the computational effort by only adding degrees

of freedom to regions that require more accuracy, we consider adaptive local refinement using hierarchical B-splines and a multi-level estimator with the maximum strategy (Garau and Vázquez, 2018). For the European Plate, we compare the results of using a uniform mesh with $16 \times 16$ elements and a hierarchical mesh arising from adaptive local refinement in Fig. 10b (right) and Fig. 10c, respectively.

### 5.2    Parameter estimation from available data

The following parameters of the model have been estimated using the method in Sect. 4 and available Mohorovičić depth map of Europe: effective elastic thickness of the lithosphere, rock density in the crust, and existing topographic load. The depth data stem from the work of Grad et al. (2009) and can be seen in Fig. 10a (right). A homogeneous effective elastic thickness



of $16\,\mathrm{km}$ and a homogeneous reference density of $2.67\,\mathrm{g\,cm^{-3}}$ are assumed, when they are not subject to estimation. These default values are furthermore used as initial values for the estimation process.

We use the Earth2014 data by Hirt and Rexer (2015) for the topographic load in Europe (see Fig. 10a, left). When topographic load is the sought parameter, its initial value is set to $1\,\mathrm{km}$ everywhere. The lithospheric depression that results from the default values and the topographic data are depicted in Fig. 10b (right). It differs from the available Mohorovičić depth data due to simplified assumptions and missing information on position-dependent parameters of the model.

The estimated effective elastic thickness is mostly around $16\,\mathrm{km}$ (see Fig. 11a, left). There are particular spots scattered 480 around the Mediterranean Sea and west of the British Isles that exhibit a slightly higher and lower thickness. A change in the effective elastic thickness of this magnitude does not significantly alter the resulting lithospheric depression; compare Fig. 11a (right) and Fig. 10b (right).

The parameter estimation predicts a higher reference density in the Baltic Shield and a lower reference density around oceans, especially in the Norwegian Sea (see Fig. 11b, left). The resulting lithospheric depression (Fig. 11b, right) is similar 485 to the Mohorovičić depth map in Fig. 10a (right). A density distribution like the estimated one can explain the observed Mohorovičić depth data well.

The lithospheric depression that results from topographic load estimation is similar to the one that results from density estimation; compare Fig. 11b (right) and Fig. 11c (right). Since the effective elastic thickness and the rock density of the lithosphere are constant, the estimated topographic load seems to mimic the contours of the Mohorovičić depth map (see 490 Fig. 11c, left).

## 5.3 Spherical model of the lithosphere

For the discretization of the variational problem in Sect. 2.4, we use a $\mathrm{C}^1$ multi-patch parametrization arising from a quad sphere projection (see Fig. 12b). The parametrization is not analysis-suitable $\mathrm{G}^1$ continuous. However, a similar one that is analysis-suitable can be constructed from it, according to Kapl et al. (2018). The new parametrization will not necessarily represent 495 the same geometry as before. Nevertheless, it can be used to obtain an analysis-suitable $\mathrm{G}^1$ multi-patch parametrization of a surface that is close to a sphere.

An effective elastic thickness of $16\,\mathrm{km}$ and $1000\,\mathrm{km}$ is chosen for the lithosphere. The latter serves to demonstrate the effects of flexural rigidity on a spherical shell under internal pressure, since the effects are negligible if the thickness is extremely small relative to the scale of the Earth. The Earth2014 data by Hirt and Rexer (2015) are mapped onto the sphere using a reverse 500 geographic projection (see Fig. 12a). The resulting deformation of the lithosphere in isostatic equilibrium is shown in Figs. 12c and 12d, where elevation refers to the radial displacement relative to the reference sphere when a spherical Earth of constant radius is assumed. Note that the scale of the coordinate system is normalized to the radius of the Earth, which is specified in Table 2.



## 6   Conclusions

In this paper, we modeled Earth's lithosphere as a thin elastic shell and presented numerical methods of isogeometric analysis
to simulate its deformation in isostatic equilibrium. Partial differential equations that involve higher-order derivatives and
require a certain smoothness of the solutions can be discretized and solved without much effort and with less degrees of
freedoms than standard finite element methods using isogeometric analysis on a single patch. For more complex geometries
that admit an analysis-suitable $G^1$ multi-patch parametrization, it is possible to construct multi-patch isogeometric spline

spaces that preserve the global $C^1$ condition. Another feature of isogeometric analysis is its ability to represent curved domains
exactly, which has been demonstrated by the simulation of a spherical shell, used to model the entire lithosphere of the Earth.
Isogeometric analysis provides a versatile tool for numerically solving problems in geoscientific applications.

Aside from simulations of the lithospheric depression on selected geographic locations, we presented a method based on
least-squares estimation constrained by partial differential equations to identify parameters that are most plausible for the plate

model when a ground truth is available. It has been applied to estimate the spatial distribution of the effective elastic thickness,
the rock density, and the topographic load of the European Plate.

*Code availability.*   The software used to compute the numerical solutions has been written in MATLAB (The MathWorks Inc., 2023) and is
available at https://doi.org/10.5281/zenodo.10950313 (Rosandi, 2024). It utilizes the GeoPDEs package (Vázquez, 2016) for isogeometric
analysis.





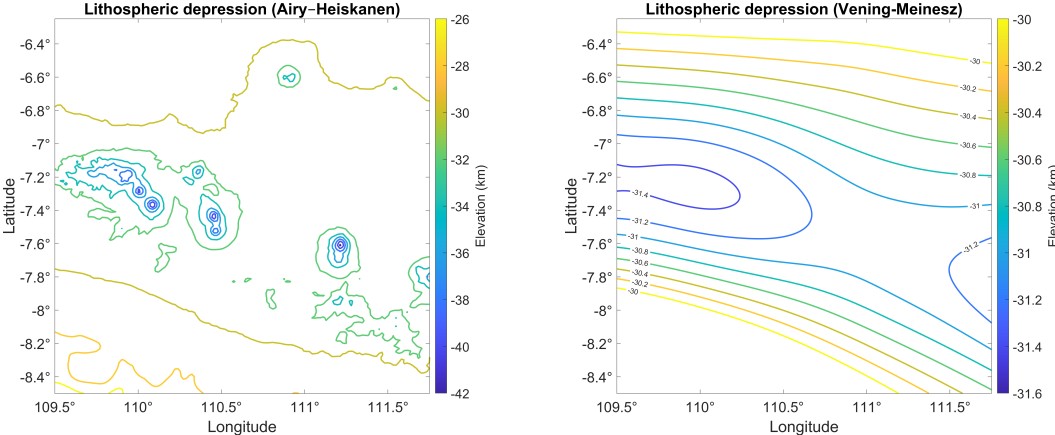

(a) Topographic map (left), corresponding load (right), and an example of a multi-patch geometry of Central Java (grid lines).

(b) Lithospheric depression in Central Java according to the Airy–Heiskanen (left) and Vening-Meinesz (right) model.

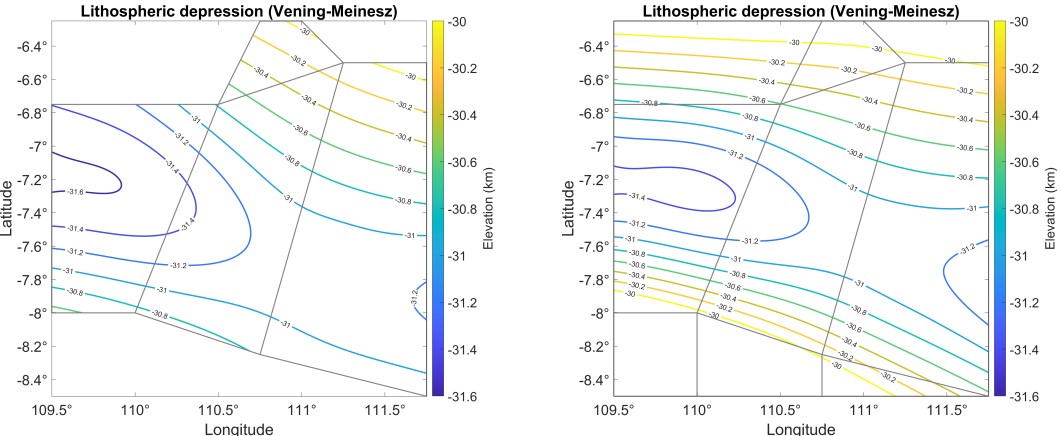

(c) Comparison between the partial (left) and the full (right) multi-patch parametrization of the domain.

**Figure 8.** Numerical simulations of the lithosphere in Central Java.



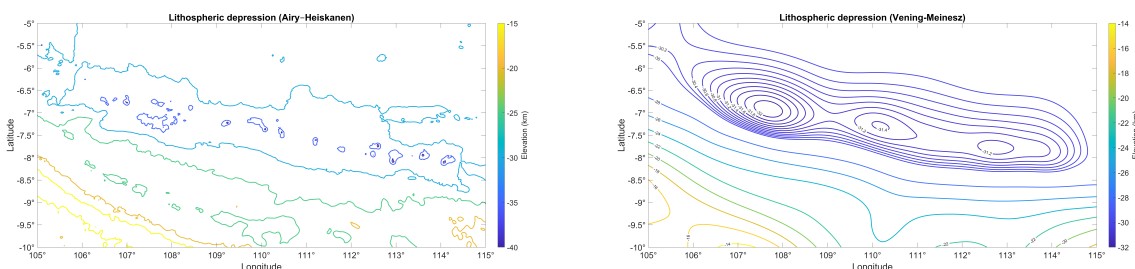

(a) Lithospheric depression in the Java Island according to the Airy–Heiskanen (left) and Vening-Meinesz (right) model.

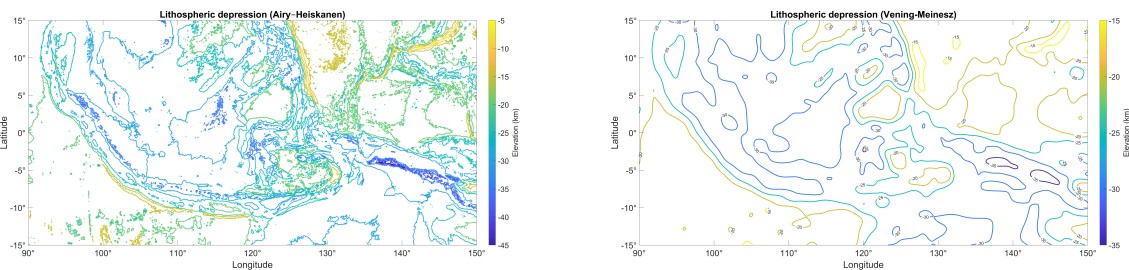

(b) Lithospheric depression in Indonesia according to the Airy–Heiskanen (left) and Vening-Meinesz (right) model.

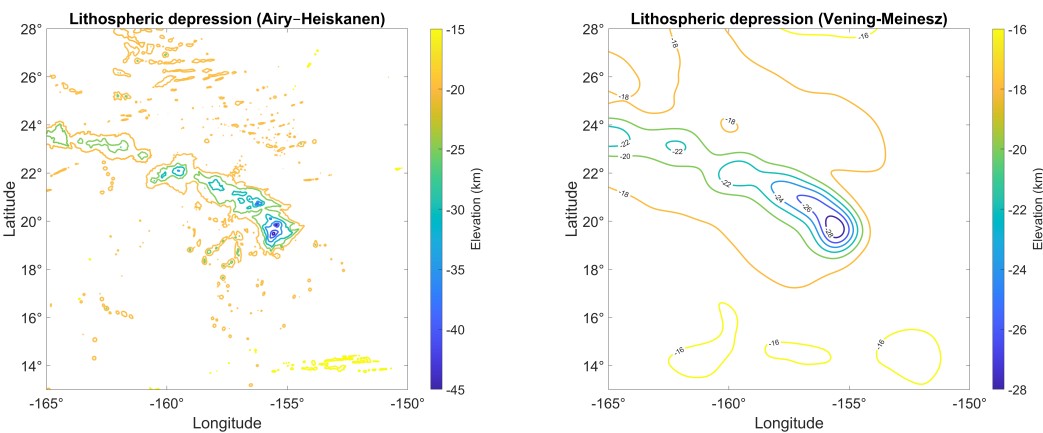

(c) Lithospheric depression in the Hawaiian Islands according to the Airy–Heiskanen (left) and Vening-Meinesz (right) model.

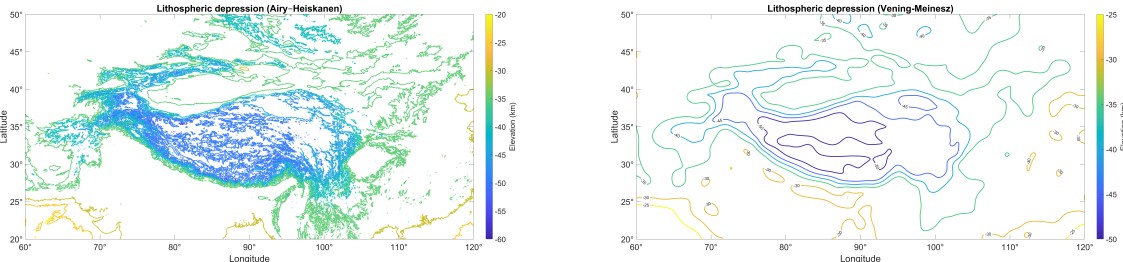

(d) Lithospheric depression in the Himalayas according to the Airy–Heiskanen (left) and Vening-Meinesz (right) model.

**Figure 9.** Numerical simulations of the lithosphere in selected geographic locations.



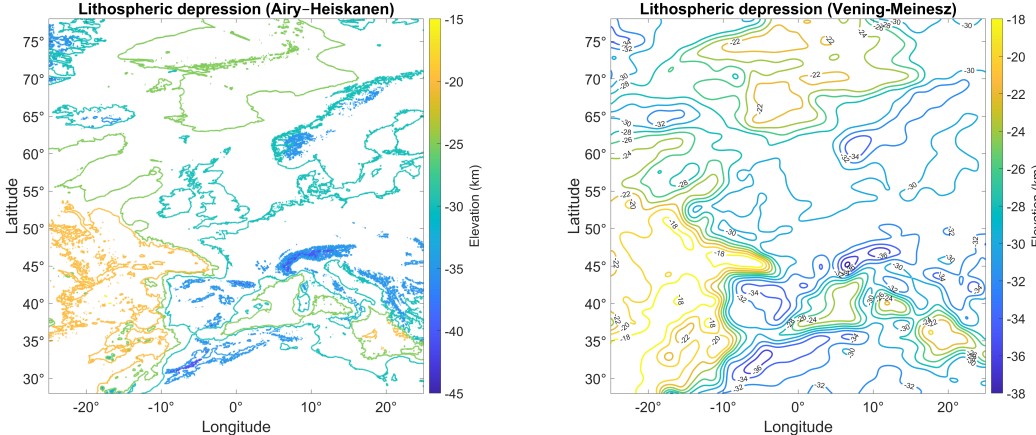

(a) Topographic load (left) and Mohorovičić depth map (right) of Europe.

(b) Lithospheric depression in Europe according to the Airy–Heiskanen (left) and Vening-Meinesz (right) model.

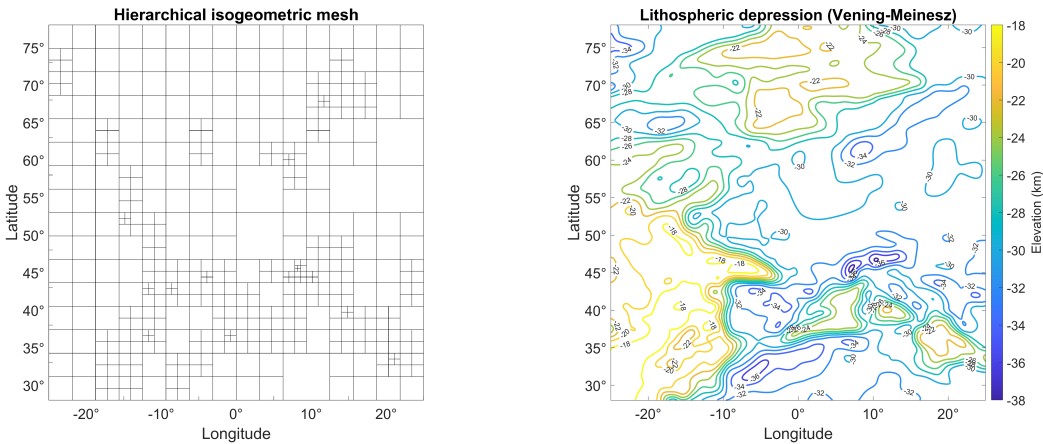

(c) Adaptive local refinement of the isogeometric mesh in Europe (left) and corresponding lithospheric depression (right).

**Figure 10.** Numerical simulations of the lithosphere in Europe.





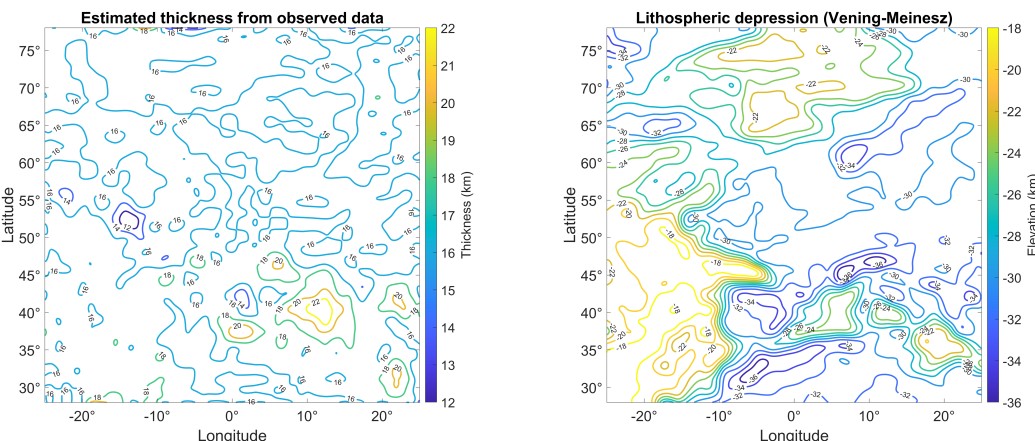

(a) Parameter estimation of the effective elastic thickness in Europe (left) and corresponding lithospheric depression (right).

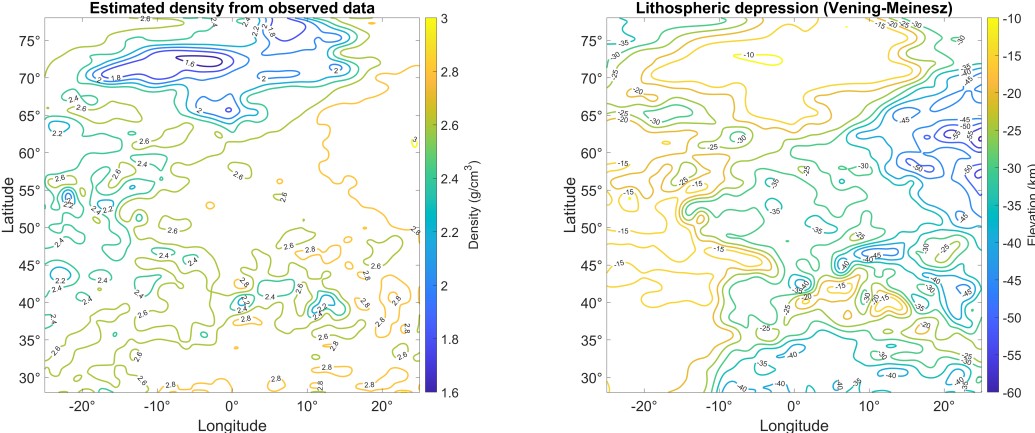

(b) Parameter estimation of the reference rock density in Europe (left) and corresponding lithospheric depression (right).

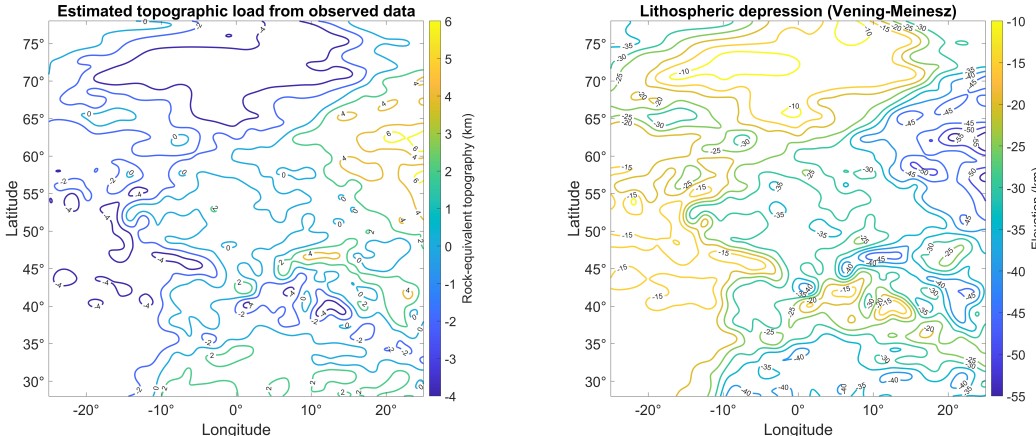

(c) Parameter estimation of the topographic load in Europe (left) and corresponding lithospheric depression (right).

**Figure 11.** Parameter estimation of the effective elastic thickness, reference rock density, and topographic load in Europe.



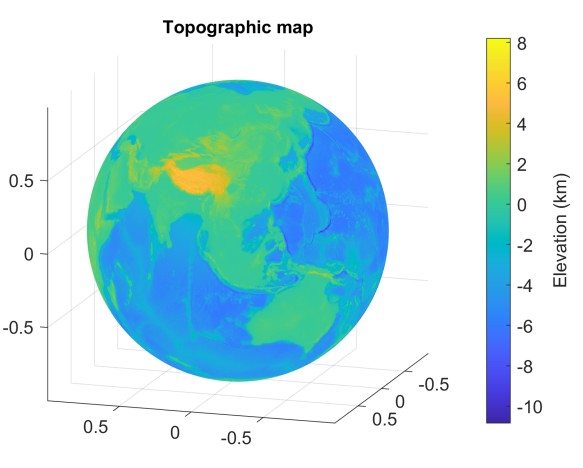

(a) Global topographic map of the spherical Earth.

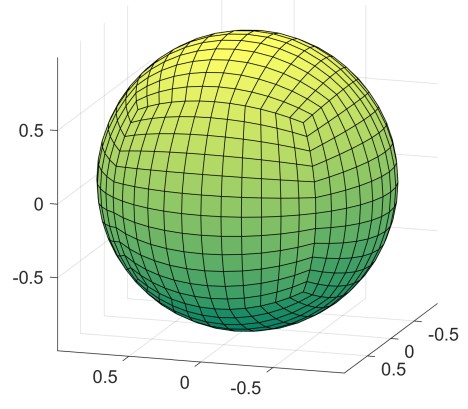

(b) Isogeometric mesh of the spherical domain.

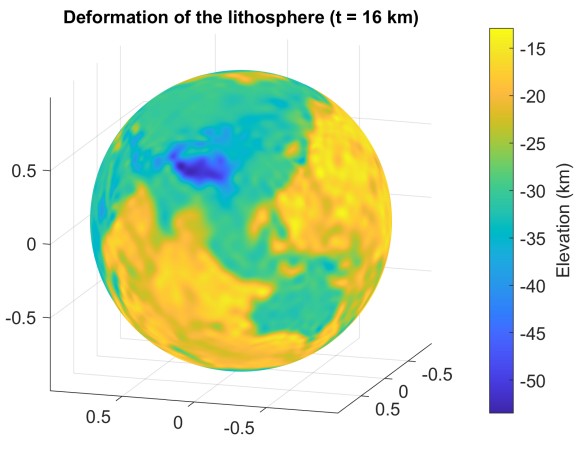

(c) Deformation of the lithosphere (t = 16 km).

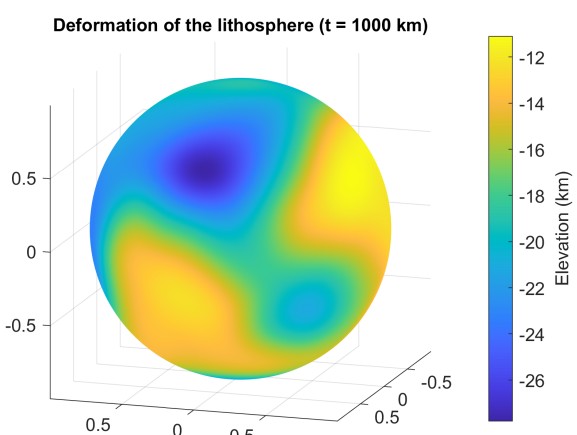

(d) Deformation of the lithosphere (t = 1000 km).

**Figure 12.** Global numerical simulations of Earth's lithosphere modeled as a thin elastic spherical shell.



## Appendix A:  List of symbols

| Symbol | Description | Symbol | Description |
|---|---|---|---|
| $\mathcal{B}$ | three-dimensional shell body | $\mathbb{R}^k$ | Euclidean space of dimension $k$ |
| $\mathcal{A}$ | reference surface (mid-surface) | $\boldsymbol{e}_0,\dots,\boldsymbol{e}_k$ | standard basis vectors in $\mathbb{R}^k$ |
| $\boldsymbol{X}$ | particle in the body | $[a,b]$ | closed interval from $a$ to $b$ |
| $\boldsymbol{\xi},\boldsymbol{\xi}_0$ | (initial) shell configuration | $(a,b)$ | open interval from $a$ to $b$ |
| $\boldsymbol{\gamma},\boldsymbol{\gamma}_0$ | (initial) mid-surface configuration | $[a,b)$ | left-closed and right-open interval from $a$ to $b$ |
| $\boldsymbol{\eta},\boldsymbol{\eta}_0$ | (initial) director field | $(a,b]$ | left-open and right-closed interval from $a$ to $b$ |
| $\boldsymbol{n},\boldsymbol{n}_0$ | (initial) unit normal vector field | $\Omega,\hat{\Omega}$ | physical and parameter domain |
| $\vartheta^1,\vartheta^2,\vartheta^3$ | local curvilinear coordinates $\boldsymbol{\vartheta}$ | $\mathcal{V},\hat{\mathcal{V}}$ | solution space on $\Omega$ and $\hat{\Omega}$ |
| $\zeta$ | thickness parameter $\vartheta^3$ | $\hat{\mathcal{V}}_{h,p}$ | discrete solution space for $\hat{\boldsymbol{u}}_{h,p}$ and $\hat{\boldsymbol{v}}_{h,p}$ |
| $t^-,t^+$ | lower and upper bound for $\zeta$ | $\boldsymbol{G}$ | degrees of freedom for the geometry function |
| $t$ | effective elastic shell thickness | $\boldsymbol{U},\boldsymbol{V}$ | degrees of freedom for trial and test functions |
| $\mathcal{D}_{[t^-,t^+]}$ | shell parameter domain | $\boldsymbol{A},\boldsymbol{L}$ | coefficient matrix and right-hand side |
| $\mathcal{D}$ | mid-surface parameter domain | $\boldsymbol{C}$ | constraint matrix for the $\mathrm{C}^1$ condition |
| $V$ | shell potential energy | $\Gamma$ | patch interface |
| $W$ | stored energy density function | $J$ | objective function |
| $\boldsymbol{F}_\mathrm{ext},f_\mathrm{ext}$ | (vertical) external body force | $I$ | reduced cost functional |
| $\boldsymbol{G}_\mathrm{ext},g_\mathrm{ext}$ | (vertical) external surface force | $R$ | state equation operator |
| $f_\mathrm{grav},f_\mathrm{buoy}$ | gravitational and buoyancy force | $q$ | design variable (flexural parameter) |
| $\boldsymbol{E}$ | Green–St.-Venant material strain tensor | $w$ | state variable (vertical deflection) |
| $\boldsymbol{E}_\mathrm{m},\boldsymbol{e}_\mathrm{m}$ | (linearized) membrane strain | $w_\mathrm{d}$ | observed data |
| $\boldsymbol{E}_\mathrm{b},\boldsymbol{e}_\mathrm{b}$ | (linearized) bending strain | $p$ | crustal depth-to-height ratio |
| $\boldsymbol{S}_\mathrm{m},\boldsymbol{s}_\mathrm{m}$ | (linearized) membrane force | $R_\mathrm{E}$ | Earth radius |
| $\boldsymbol{S}_\mathrm{b},\boldsymbol{s}_\mathrm{b}$ | (linearized) bending moment | $\mathrm{C}^1$ | space of continuously differentiable functions |
| $\boldsymbol{K}$ | elasticity tensor | $\mathrm{G}^1$ | space of geometrically $\mathrm{C}^1$ functions |
| $E$ | Young's modulus | $\mathrm{L}^2$ | Lebesgue space of square-integrable functions |
| $\nu$ | Poisson's ratio | $\mathrm{H}^2$ | Hilbert–Sobolev space of order 2 |
| $D,\widetilde{D}$ | flexural rigidity (for a beam) | $\lvert\cdot\rvert$ | absolute value |
| $\boldsymbol{u}$ | displacement field | $\lVert\cdot\rVert$ | Euclidean norm |
| $\boldsymbol{v}$ | variation of the displacement | $\langle\cdot,\cdot\rangle$ | duality pairing |
| $w$ | vertical deflection field | $\cdot$ | standard dot product |
| $v$ | variation of the vertical deflection | $:$ | double tensor contraction |
| $a$ | bilinear form for the stiffness matrix | $\times$ | cross product |
| $b$ | bilinear form for the mass matrix | $\otimes$ | tensor product |
| $c$ | linear functional for gravitational load | $[\![\cdot]\!]$ | interface jump |
| $\ell$ | linear functional for external load | $\mathrm{dA}$ | area element |
| $d,d_0$ | (initial) mid-surface depth | $\mathrm{dS}$ | length element |
| $t_0$ | standard crustal thickness | $\mathrm{d}x,\mathrm{d}y,\mathrm{d}z$ | differential of coordinate functions $x,y,z$ |
| $h$ | topographic elevation | $\frac{\mathrm{d}}{\mathrm{d}x},\frac{\mathrm{d}^2}{\mathrm{d}x^2}$ | first and second derivative operator |
| $r$ | rock-equivalent topography | $\partial$ | boundary operator |
| $\varrho$ | density of overlying mass | $\partial_1,\partial_2$ | partial differential operators |
| $\varrho_\mathrm{m}$ | upper mantle density | $\delta$ | first variation operator |
| $\varrho_\mathrm{r}$ | reference rock density | $\nabla$ | gradient operator |
| $g$ | gravitational acceleration | $\nabla^2$ | Hessian operator |
| $B_{k,p}$ | $k$-th B-spline basis function of degree $p$ | $\Delta$ | Laplace operator |
| $N_{k,p}^{\boldsymbol{\omega}}$ | NURBS basis function with weights $\boldsymbol{\omega}$ | $\mathcal{O}$ | big O symbol |
| $\boldsymbol{N}_\alpha,\boldsymbol{N}_{k,l}$ | $\alpha$-th vector-valued NURBS basis function | $\in$ | element symbol |
| $\mathcal{S}_{d,r}^{\boldsymbol{\omega}}(\boldsymbol{\Theta},\boldsymbol{p})$ | space of $d$-variate NURBS patches in $\mathbb{R}^r$ | $\subset$ | subset symbol |
| $\vartheta_0,\dots,\vartheta_m$ | spline knots in a knot sequence $\boldsymbol{\Theta}$ | $\rightarrow$ | mapping symbol |
| $p_1,\dots,p_d$ | spline degrees in a tuple of degrees $\boldsymbol{p}$ | $\circ$ | function composition |
| $\boldsymbol{c}_0,\dots,\boldsymbol{c}_n$ | control points of a NURBS patch in $\mathbb{R}^r$ | $\top$ | matrix transpose |
| $\omega_0,\dots,\omega_n$ | NURBS weights in a tuple of weights $\boldsymbol{\omega}$ | $\vert$ | restriction symbol |
| $\omega$ | weight function | | |



*Author contributions.* YR designed the study and put forward the geophysical problem to be investigated. RR formulated the mathematical models and numerical methods to solve the problem, collected data, and developed the software. The numerical simulations and data analysis were carried out by RR. The first draft was written by RR and YR. BS evaluated the draft and finalized the report. All authors commented on the manuscript, reviewed the results, and approved the final version of the manuscript.

*Competing interests.* The authors declare that there are no competing interests.

*Acknowledgements.* Y. Rosandi is grateful for the Academic Leadership Grant, Universitas Padjadjaran, Bandung, Indonesia, contract No. 1549/UN6.3.1/PT.00/2023. The authors acknowledge the personal discussion with W. Suryanto, Universitas Gadjah Mada, Yogyakarta, Indonesia.



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

Methods in Applied Mechanics and Engineering, 411, 2023.

Garau, E. M. and Vázquez, R.: Algorithms for the implementation of adaptive isogeometric methods using hierarchical B-splines, Applied
Numerical Mathematics, 123, 57–78, 2018.

Grad, M., Tiira, T., and Group, E. W.: The Moho depth map of the European Plate, Geophysical Journal International, 176, 279–292, 2009.

Gutenberg, B.: Isostasy and its meaning, Tellus, 1, 1–5, 1949.

Hinze, M., Pinnau, R., Ulbrich, M., and Ulbrich, S.: Optimization with PDE Constraints, Mathematical Modelling: Theory and Applications,
Springer, 2008.

Hirt, C. and Rexer, M.: Earth2014: 1 arc-min shape, topography, bedrock and ice-sheet models – available as gridded data and degree-10,800
spherical harmonics, International Journal of Applied Earth Observation and Geoinformation, 39, 103–112, 2015.

Hughes, T. J. R., Cottrell, J. A., and Bazilevs, Y.: Isogeometric analysis: CAD, finite elements, NURBS, exact geometry and mesh refinement,
Computer Methods in Applied Mechanics and Engineering, 194, 4135–4195, 2005.

Jüttler, B. and Simeon, B., eds.: Isogeometric Analysis and Applications 2014, Lecture Notes in Computational Science and Engineering,
Springer, 2015.



Kapl, M., Sangalli, G., and Takacs, T.: Construction of analysis-suitable $G^1$ planar multi-patch parameterizations, Computer-Aided Design, 97, 41–55, 2018.

Kiendl, J., Bletzinger, K.-U., Linhard, J., and Wüchner, R.: Isogeometric shell analysis with Kirchhoff–Love elements, Computer Methods in Applied Mechanics and Engineering, 198, 3902–3914, 2009.

Koiter, W. T.: On the nonlinear theory of thin elastic shells, Proc. K. Ned. Akad. Wet., B 69, 1–54, 1966.

Lowrie, W.: Fundamentals of Geophysics, Cambridge University Press, 1997.

Lyche, T., Manni, C., and Speleers, H., eds.: Splines and PDEs: From Approximation Theory to Numerical Linear Algebra, Springer, 2018.

Manríquez, P., Contreras-Reyes, E., and Osses, A.: Lithospheric 3-D flexure modelling of the oceanic plate seaward of the trench using variable elastic thickness, Geophysical Journal International, 196, 681–693, 2014.

Marsden, J. E. and Hughes, T. J. R.: Mathematical Foundations of Elasticity, Dover Civil and Mechanical Engineering Series, 1994.

Melosh, H. J.: Planetary Surface Processes, Cambridge Planetary Science, 2011.

Mostafa, M., Sivaselvan, M. V., and Felippa, C. A.: A solid-shell corotational element based on ANDES, ANS and EAS for geometrically nonlinear structural analysis, International Journal for Numerical Methods in Engineering, 95, 145–180, 2013.

Neunteufel, M. and Schöberl, J.: The Hellan–Herrmann–Johnson method for nonlinear shells, Computers and Structures, 225, 2019.

Nunn, J. A. and Aires, J. R.: Gravity anomalies and flexure of the lithosphere at the Middle Amazon Basin, Brazil, Journal of Geophysical
Research, 93, 415–428, 1988.

Oñate, E. and Zárate, F.: Rotation-free triangular plate and shell elements, International Journal for Numerical Methods in Engineering, 47, 557–603, 2000.

Pelletier, J. D.: Quantitative Modeling of Earth Surface Processes, Cambridge University Press, 2008.

Piegl, L. A. and Tiller, W.: The NURBS Book, Springer, 1995.

Rogers, N., ed.: An Introduction to Our Dynamic Planet, Cambridge University Press, 2008.

Rosandi, R.: rozanxt/igalith: Igalith (version 1.0.0), https://doi.org/10.5281/zenodo.10950313, 2024.

Steigmann, D.: Koiter's shell theory from the perspective of three-dimensional nonlinear elasticity, Journal of Elasticity, 111, 91–107, 2013.

The MathWorks Inc.: MATLAB version: 23.2.0.2365128 (R2023b), 2023.

van Brummelen, H., Vuik, C., Möller, M., Verhoosel, C., Simeon, B., and Jüttler, B., eds.: Isogeometric Analysis and Applications 2018,
Lecture Notes in Computational Science and Engineering, Springer, 2021.

Vázquez, R.: A new design for the implementation of isogeometric analysis in Octave and Matlab: GeoPDEs 3.0, Computers and Mathematics with Applications, 72, 523–554, 2016.

Vening-Meinesz, F. A.: Une nouvelle methode pour la reduction isostatique régionale de l'intensité de la pesanteur, Bulletin Géodésique, 29, 33–51, 1931.

Vuong, A.-V., Giannelli, C., Jüttler, B., and Simeon, B.: A hierarchical approach to adaptive local refinement in isogeometric analysis, Computer Methods in Applied Mechanics and Engineering, 200, 3554–3567, 2011.

Watts, A. B.: Isostasy and Flexure of the Lithosphere, Cambridge University Press, 2001.

Wickert, A. D.: Open-source modular solutions for flexural isostasy, Geoscientific Model Development, 9, 997–1017, 2016.

Zienkiewicz, O. C., Taylor, R. L., and Zhu, J. Z.: The Finite Element Method: Its Basis and Fundamentals, Butterworth–Heinemann, 2013.