# Peer review of "Isogeometric analysis of the lithosphere under topographic loading: Igalith v1.0.0"

_EGUsphere, 2024_

## Author Comment (AC1)

**Well-posedness of the isostatic boundary value problem**

In the following, we use the notation from the preprint [4]. Without loss of generality, the coefficients in the bilinear forms

$$a(w, v) = \int_{\mathcal{A}} D\big(\nu \Delta w \Delta v + (1 - \nu) \nabla^2 w : \nabla^2 v\big) \, \mathrm{dA},$$
$$b(w, v) = \int_{\mathcal{A}} (\varrho_{\mathrm{m}} - \varrho_{\mathrm{r}}) g w v \, \mathrm{dA}, \tag{1}$$

can be assumed to be real numbers with $D > 0$, $0 \le \nu \le 0.5$, $g > 0$, and $\varrho_{\mathrm{m}} > \varrho_{\mathrm{r}}$. If the coefficients are variable in space, we only require that $D(1 - \nu)$ and $(\varrho_{\mathrm{m}} - \varrho_{\mathrm{r}})g$ are bounded from below by a positive number.

**Claim:** Let $\mathcal{A} \subset \mathbb{R}^2$ be a bounded Lipschitz domain. Then there exists a constant $\alpha > 0$ such that

$$a(w, w) + b(w, w) \ge \alpha \|w\|_{\mathrm{H}^2(\mathcal{A})}^2 \tag{2}$$

for all $w \in \mathrm{H}^2(\mathcal{A})$.

**Proof:** Let $\mathrm{C}^\infty(\mathcal{A})$ denote the space of smooth real-valued functions on $\mathcal{A}$. Since $\mathrm{C}^\infty(\mathcal{A})$ is dense in $\mathrm{H}^2(\mathcal{A})$ with respect to the $\mathrm{H}^2$ norm, it suffices to establish the inequality for $w \in \mathrm{C}^\infty(\mathcal{A})$. By definition of the bilinear forms, we have that

$$\begin{aligned}
a(w, w) + b(w, w) &= \int_{\mathcal{A}} D\big(\nu |\Delta w|^2 + (1 - \nu) |\nabla^2 w|^2\big) \, \mathrm{dA} + \int_{\mathcal{A}} (\varrho_{\mathrm{m}} - \varrho_{\mathrm{r}}) g |w|^2 \, \mathrm{dA} \\
&\ge D(1 - \nu) \|\nabla^2 w\|_{\mathrm{L}^2(\mathcal{A})}^2 + (\varrho_{\mathrm{m}} - \varrho_{\mathrm{r}}) g \|w\|_{\mathrm{L}^2(\mathcal{A})}^2 \\
&\ge C \left( \|\nabla^2 w\|_{\mathrm{L}^2(\mathcal{A})}^2 + \|w\|_{\mathrm{L}^2(\mathcal{A})}^2 \right)
\end{aligned} \tag{3}$$

with $C = \min\{D(1 - \nu), (\varrho_{\mathrm{m}} - \varrho_{\mathrm{r}})g\} > 0$. According to the Ehrling–Gagliardo–Nirenberg interpolation inequality in [1, Theorem 5.2] or the equivalence of norms in [3, Theorem 1.8], there exists a constant $K > 0$ such that

$$\|\nabla w\|_{\mathrm{L}^2(\mathcal{A})}^2 \le K \left( \|\nabla^2 w\|_{\mathrm{L}^2(\mathcal{A})}^2 + \|w\|_{\mathrm{L}^2(\mathcal{A})}^2 \right) \tag{4}$$

in the case of a bounded Lipschitz domain $\mathcal{A}$. Applying (4) to (3) yields

$$\begin{aligned}
a(w, w) + b(w, w) &\ge \frac{C}{2} \left( \|\nabla^2 w\|_{\mathrm{L}^2(\mathcal{A})}^2 + \|w\|_{\mathrm{L}^2(\mathcal{A})}^2 \right) + \frac{C}{2K} \|\nabla w\|_{\mathrm{L}^2(\mathcal{A})}^2 \\
&\ge \alpha \|w\|_{\mathrm{H}^2(\mathcal{A})}^2
\end{aligned} \tag{5}$$

with the coercivity constant $\alpha = \min\{C/2, C/(2K)\} > 0$. $\qquad \square$

The above shows that $a + b \colon \mathrm{H}^2(\mathcal{A}) \times \mathrm{H}^2(\mathcal{A}) \to \mathbb{R}$ is a coercive bilinear form. Furthermore, symmetry and continuity of $a + b$ are easily verified. The well-posedness of the isostatic boundary value problem with Neumann boundary conditions then follows from a standard argument using the Lax–Milgram theorem [2, Chapter II, Section 2–3].

**References**

[1] Robert A. Adams and John J. F. Fournier. *Sobolev Spaces.* Academic Press, 2003.

[2] Dietrich Braess. *Finite Elements: Theory, Fast Solvers, and Applications in Solid Mechanics.* Cambridge University Press, 2007.

[3] Jindřich Nečas. *Direct Methods in the Theory of Elliptic Equations.* Springer, 2012.

[4] Rozan Rosandi, Yudi Rosandi, and Bernd Simeon. Isogeometric analysis of the lithosphere under topographic loading: Igalith v1.0.0. *EGUsphere [preprint]*, 2024.

---

## Author Response (AR1)

**Author's response to the reviews and relevant changes to the manuscript egusphere-2024-1093**

This document provides a detailed point-by-point response to all referee comments and the relevant changes that have been made in the manuscript. Referee comments are shown in *italics*. Author's responses are shown in **bold**.
* * *
Anonymous Referee #1:

**We would like to thank Anonymous Referee #1 for their careful reading and the highly detailed comments as well as useful suggestions to our manuscript.**

*This paper presents the applicability of Isogeometric Analysis for problems in geoscience. More particularly, it is herein considered the modelling of the lithosphere as a thin shell subject to gravitational and buoyancy forces exerted by the topographic features above the lithosphere and the upper layers of the athenosphere, respectively. The paper is overall very well written, easy to read, and the findings are accompanied/supported by numerical evidence. Nevertheless, I would like to have the remarks below addressed in an adequate manner before I can suggest publication of the paper:*

*1. On page 2 the authors mention "Further features and capabilities of isogeometric analysis presented in this paper are the exact representation of curved domains, the coupling of multiple patches." The coupling of multiple patches is not a feature of IGA but rather a challenge of the method, from which standard finite element methods do not suffer. I would not enlist the coupling of multiple patches as a feature, but I would highlight that patch coupling is essential in IGA for problems of practical relevance.*

**We have omitted the coupling of multiple patches from the list of features of isogeometric analysis and instead highlighted its importance in problems of practical relevance as well as the research efforts that have been undertaken.**

*2. On page 5 the auhors provide the parametric description of the continuum using contravariant coordinates. They refer to these as curvilinear coordinates in general. Since the superscript is used, I would recommend calling these as contravariant and not simply curvilinear coordinates.*

**It is common in differential geometry to denote curvilinear coordinates using superscripts. Contravariant coordinates usually refer to the coordinates of a vector field on some manifold with respect to the basis vectors induced by a coordinate system on the manifold, which form a coordinate frame on the tangent bundle and differ from the original coordinate system. We would like to keep using the term "curvilinear coordinates" to refer to the components of the coordinate chart theta, which has also been done in Kiendl et al. (2009): "Isogeometric shell analysis with Kirchhoff–Love elements." Computer Methods in Applied Mechanics and Engineering 198(49–52):3902–3914.**

*3. On Section 2.1.1 Shell Configuration the authors expland in very detail in the variational formulation of the Kirchhoff–Love shell. These equations can be found in many resources, see for instance in Kiendl, Josef, et al. "Isogeometric shell analysis with Kirchhoff–Love elements." Computer methods in applied mechanics and engineering 198.49–52 (2009): 3902–3914. I would recommend reducing this part to the absolute minimum and just citing an appropriate resource.*

**The section "Shell configuration" has been removed and replaced with a paragraph that contains references on the kinematic assumptions and the minimum amount of notions required for the variational formulation of the shell equations.**

*4. On page 7 the authors mention the following: "In the one-dimensional case with $w = w(x)$, the bending term reduces further, which leads to a fourth-order differential equation for an Euler–Bernoulli beam when considering the strong formulation of the problem without boundary conditions." I think this is an oversimplification of what is in fact going on. The mathematical and mechanical effect on reducing a structural element from a 3d continuum to a beam introduces many additional mechanical effects (torsion, bending, twisting, etc.) that can not be understood as a reduction from a plate (or shell). I do not see a reason why this section is added and I would recommend removing it (see Bauer, A. M., et al. "Nonlinear isogeometric spatial Bernoulli beam." Computer Methods in Applied Mechanics and Engineering 303 (2016): 101–127. for more information).*

**The purpose of the paragraph was to establish a relation to the commonly used one-dimensional model for flexural isostasy, e.g., in Chapter 5 of Pelletier (2008). As opposed to a general spatial beam model, the initial undeformed configuration is assumed to be straight and only vertical deflections are considered, thus ignoring the mechanical effects of torsion and twisting. We have followed the recommendation to remove the paragraph.**

*5. On page 7 the authors mention "We model the lithosphere as a thin elastic plate." Is this a thin elastic plate or a thin shell? The equations the authors laid down in Section 2.1.1 refer to a shell.*

**Both a shell and a plate model are considered in our work. In Section 2.2 and 2.3, we use the plate model described in Section 2.1.3. The topographic loading and buoyant forces are adapted to the shell model in Section 2.4.**

*6. On page 7 the authors refer to a "Kirchhoff–Love plate". To my knowledge "Kirchhoff–Love" is the designation for a shell, the corresponding plate is called just "Kirchhoff".*

**"Kirchhoff–Love" is another common designation for the classical plate theory based on the hypotheses of Kirchhoff and Love, which is used for both plates and shells in the literature. Since this might be a misattribution, because Love developed a theory of shells rather than plates in his seminal work Love (1888): "The Small Free Vibrations and Deformation of a Thin Elastic Shell." Philosophical Transactions of the Royal Society of London A 179:491–546, we have replaced instances of "Kirchhoff–Love" with just "Kirchhoff" in the context of plates.**

*7. On page 8 the authors write the following "Instead of working with the actual topographic elevation, we use a mass representation r obtained by taking the mass above the mid-surface of the lithosphere and normalizing it by some depth-independent reference density $\varrho_\mathrm{r}$. We choose the reference density as the mean rock density from the current depth of the mid-surface to its initial depth and assume that it is homogeneous." This approach is not clear to me. I would like to ask the authors to add some reference if it is standard in the literature? I am not sure what this modelling implies.*

**The mass representation is termed "rock-equivalent topography". A more detailed description can be found in the appendix of Hirt and Rexer (2015).**

*8. On page 9 the authors mention the following: "For the full Neumann problem, it can*

*be shown using Korn's inequality that the variational problem is well-posed, provided that A is a bounded domain with piecewise smooth boundary and the coefficient $(\varrho_{\mathrm{m}} - \varrho_{\mathrm{r}})g$ in the buoyancy term is bounded from below by a positive number." Is there any reference for this statement? When the authors mention that something can be shown it is advisable to either add a reference or a proof, if there is no reference.*

**The sentence has been rephrased. Unfortunately, we did not find any reference that covers this particular problem and realized that Korn's inequality, which was used to prove the well-posedness of the plate problem in Braess (2007), does not play a role in our case. However, the well-posedness of the isostatic boundary value problem with Neumann boundary conditions follows directly from a standard argument using the Lax–Milgram theorem (see, e.g., Chapter II, Section 2–3 in Braess (2007)) and the fact that the $\mathbf{H^2}$ norm is equivalent to the one induced by the bilinear form $\mathbf{a(w, v) + b(w, v)}$ on a bounded Lipschitz domain (see Theorem 1.8 in Nečas (2012): "Direct Methods in the Theory of Elliptic Equations." Springer. or the Ehrling–Nirenberg–Gagliardo interpolation inequality in Theorem 5.2 of Adams and Fournier (2003): "Sobolev Spaces." Academic Press.), which is used to show the $\mathbf{H^2}$ coercivity of the bilinear form. We provided a document with a proof of the well-posedness in the supplement, but we would rather omit it from the final version of the manuscript, since the proof is elementary and not the primary focus of our paper.**

*9. On pages 9–10 the authors mention the following: "From a modeling point of view, the results may not reflect the physical reality since the Earth consists of different regimes and tectonic plates that interact with each other in a complex manner. Furthermore, due to the large scale of the simulation, the effects of flexural rigidity will not be visible. Nevertheless, we assume that the entire lithosphere can be modeled as a single spherical shell to showcase the capabilities of isogeometric analysis in numerical simulations on curved domains, especially on a spherical domain." Is this to be understood as an academic study without applicability to geoscience? I am not sure how to interpret this statement as it mentions that the results may not reflect the physical reality and that the study aims to showcase the capabilities of isogeometric analysis in numerical simulations on curved domains.*

**Modeling the entire lithosphere as a single spherical shell is an enormous simplification that ignores many geophysical circumstances. We want to make clear that this work is an academic study that still holds some relevance in applications. The numerical simulations yield accurate results and can be used for predictions under the assumption that the lithosphere behaves ideally as described. However, there are still a lot of phenomena that can be put into consideration to improve the realism of the model.**

*10. On page 10 the authors mention the following: "to both parametrize the domain and construct finite element approximations of solutions to the equations." I would recommend rephrasing this to "to both parametrize the domain and construct finite element approximations of solutions to the corresponding partial differential equations."*

**The sentence has been rephrased as recommended.**

*11. On page 11 the authors mention the following: "In the following, we consider splines on the unit interval $[0, 1]$ with an open knot sequence, i.e., the first and last knot values have multiplicity $p + 1$. Then the knot sequence has the form". I would like to ask the*

*authors to mention the interpolation effect that is enforced when the first and last know values have the prescribed multiplicity. Why are the authors restricting themselves to this choice and what is the implication of doing so?*

**The phrase "to enable interpolatory control points at the boundary" has been added. Using an open knot sequence is a standard way to enforce Dirichlet boundary conditions or to join multiple patches together at the boundary with $C^0$ continuity. The theory and implementation are not restricted to open knot sequences or the unit interval [0, 1]. We assume them for the sake of simplicity.**

*12. In the caption of Figure 5 the authors write "bivariate quadratic". I would recommend replacing "quadratic" with "biquadratic".*

**The term "bivariate quadratic" has been replaced with "biquadratic".**

*13. Regarding Section 3.1.2 NURBS basis functions, I feel that it is spent quite a lot of real estate on detailing the B-Spline and NURBS basis functions, information that can be found easily in many other resources. Moreover, the authors do not cite these other resources in an adequate manner, but only Piegl and Tiller. I would like to ask the authors to add more references as needed.*

**More references have been added as requested. The aim of the paper is to introduce isogeometric finite element methods using B-splines and NURBS to the geoscientific community, which is the reason why they have been presented in detail.**

*14. On page 16 the authors mention the following: "We can stack the control points on top of each other". I would recommend rephrasing this to "the control point coordinates are organized in vectors". That sound more appropriate for a scientific publication.*

**The sentence has been rephrased as follows: "The coordinates of the control points are organized in a single column vector so that the Galerkin equation can be written in matrix-vector form."**

*15. On the same page the authors mention "There are various methods that can be employed to achieve this" when referring to NURBS multipatches. There are so many publications and studies on this matter, I would like to ask the authors to add adequate references. The reference list is herein quite poor.*

**More references on multi-patch $C^1$ coupling have been added as requested with emphasis on weak coupling methods, such as penalty, Nitsche, and mortar methods, as well as strong coupling methods that are based on the construction of globally $C^1$ (or approximately $C^1$) shape functions.**

*16. On the same page the authors mention the following: "by replacing shape functions at the boundary of each patch that coincides with the boundary of another patch with interface functions that span over multiple patches." Could the authors clarify whether this is kind of a bending strip? See Kiendl et al. 2010.*

**The construction in Farahat et al. (2023) is not related to the bending strip method, which couples the shape functions on neighboring patches weakly by introducing additional terms to the variational formulation that penalize changes in the angle at the interfaces. Instead, a globally $C^1$ isogeometric spline space, which consists of patch-interior, edge, and vertex basis functions, is constructed and used to enforce the $C^1$ interface condition strongly.**

*17. On the same page the authors mention the following: "Another approach is to stitch shape functions at the patch interfaces together by imposing the $C^1$ condition." Could the authors clarify whether it is herein meant that the $C^0$ continuity is already assumed (meaning that the patches are conforming along their interface) and that the authors are just constraining the second row of control points in such a way as to achieve $C^1$ (or $G^1$) continuity? It is still unclear to me how the $C^1$ continuity is enforced (Is this a Penalty, Lagrange Multipliers, Mortar, or any other approach?).*

**The $C^0$ continuity is already assumed by using patches that share the same configuration for interpolatory control points at the interfaces. Control points that influence the derivative of solutions at patch interfaces are then constrained by computing and using the null space of the $C^1$ constraint matrix to achieve approximate $C^1$ continuity. This is the numerical implementation described in Collin et al. (2016), which corresponds to a strong coupling method with approximately $C^1$ shape functions.**

*18. On the same page the authors mention the following: "Aside from that, the computation of the null space corresponding to the $C^1$ constraint is generally a difficult task numerically." Do the authors herein mean that the coupled system becomes non-convex?*

**We do not claim that the coupled system becomes non-convex. According to Collin et al. (2016), "The numerical computation [of the null space] is a hard task for non-analysis-suitable $G^1$ geometries. Indeed, the non-zero eigenvalues [of the constraint matrix] are not well-separated from the eigenvalues that are numerically zero (close to machine precision)." We have decided to remove the sentence. The difficulty of constructing a multi-patch $C^1$ isogeometric spline space for complex geometries is still mentioned in the next paragraph.**

*19. Yet on the same page, the authors mention the following: "In this work, we will mainly consider planar domains that result from joining convex quadrilaterals along the sides. It has been shown that the class of bilinear $G^1$ parametrizations is analysis-suitable, so that optimal convergence can be achieved in this setting." I understood that the authors would model the whole Earth's lithosphere as a spherical shell. How can the domains be considered planar in such a case?*

**For the plate model, only planar domains that result from joining convex quadrilaterals along the sides have been considered in our work. For the spherical shell model, we use the multi-patch parametrization specified in Section 5.3.**

*20. On page 20 the authors mention the following: "It differs from the single-patch result due to the missing data outside of the simulation domain that are replaced by Neumann boundary conditions." I think it is natural that the results naturally differ. Could the authors clarify whether the point of this comparison is to show that one can obtain satisfactory results also when using less DOFs? If not, could they please clarify what the exact point of this discussion is?*

**The point of the discussion was not to show that one can obtain satisfactory results when using less degrees of freedom. The example in Fig. 8c (left) showcases the use of multi-patch isogeometric analysis to perform simulation on planar domains with a more complex geometry, which is useful when the data required for the simulation are only available on certain parts of Earth's surface. The example in Fig. 8c (right) serves to show that subdividing a single**

**patch into multiple ones yields a result that is similar to the original one in Fig. 8b (right).**

*21. On page 21 the authors mention the following: "When topographic load is the sought parameter, its initial value is set to 1 km." Could the authors clarify why is the value of the topographic load measured in km? km should measure distances, right?*

**We express topographic load through rock-equivalent topography, which corresponds to the height of overlying mass when a constant rock density is used. The corresponding topographic load per unit area simply differs by a factor of $\varrho_\mathrm{r} g$. The part of the sentence has been replaced with "the initial value for the corresponding rock-equivalent topography is set to 1 km" to make it clearer.**

*22. I tried reproducing the results, but I stumbled upon some difficulties with the provided open-source repositories. I am able to run script script_central_java.m provided that all necessary third-party software is installed, but I am unable to execute script script_spherical_earth.m because I receive an error message at line 156 of file op_KL_bending_stress.m from GeoPDEs, namely: "Error using .^ Invalid data type. Argument must be numeric, char, or logical." I tried to debug the issue, but it is probably due to incompatibility of the version of the GeoPDEs library with the one used to produce the results in this study. Could the authors also upload your code also on GitHub and provide clear instructions on the versions of the third-party libraries needed to reproduce the results shown in this study?*

**There is a line of code we did not realize was modified in our own version of the third-party library. As opposed to the plate model, the current implementation of the shell model in the GeoPDEs package (version 3.2.2) does not support variable thickness yet. You can either replace `t_coeff` in line 56 of the file `packages/geopdes/inst/space/@sp_vector/op_KL_shells_tp.m` with `t_coeff(x{:})` or remove `prb.th = @(x, y, z) prb.th * ones(size(x));` from line 27 of the file `isostasy/script_spherical_earth.m`. This has been updated in the GitHub repository `https://github.com/rozanxt/igalith`, which is now referenced in the manuscript.**

*23. Otherwise, the results section is really great and I am impressed with the results shown in Figure 12. Great work, congratulations!*

**Thank you very much for the insightful review!**

Link to the supplement on the well-posedness of the isostatic boundary value problem: `https://editor.copernicus.org/index.php?_mdl=msover_md&_jrl=778&_lcm=oc108lcm109w&_acm=get_comm_sup_file&_ms=119382&c=280243&salt=2082756890563503959`
* * *
Referee #2 (Tony Lowry):

**The authors thank Tony Lowry for the thorough and critical review. The comments and references are valuable for improving the quality of our manuscript.**

*This paper describes a refinement to finite element modeling of thin-plate isostasy that incorporates isogeometric representation of surfaces such as topography and Moho flexure. I am unfamiliar with the existing literature on spherical finite element approaches to modeling isostasy (though presumably there are many such), and without a head-to-head comparison of results from those other methods, it is unclear to me how much of an improvement is afforded by the isogeometric approach to isostatic modeling presented here relative to (say)*

*representing topographic and internal loads or the Moho response as stress fields acting on the plate. The requirement of fewer degrees of freedom sounds promising though (although it is not rigorously explored here), even if it is unclear that a more realistic representation of surface curvature affords other advantages.*

**The advantage of using isogeometric methods over standard C$^1$ finite element methods lies in the observation that mesh refinement in the latter will increase the number of interfaces between elements where the C$^1$ condition has to be enforced, thus leading to significantly more degrees of freedom, whereas the number of patch interfaces will remain the same when using the isogeometric approach. It is more efficient in the sense that more accurate numerical results can be obtained with less computational effort.**

*Effective elastic thickness (Te) estimation may benefit from improved finite element approaches in a couple of ways: (1) Finite-element methods are commonly used to generate synthetic data with known property variations as a means of testing inverse methods (e.g., Kirby, "Spectral methods for the estimation of the effective elastic thickness of the lithosphere"), and (2) The inverse methods currently rely on the linearity of flexural response afforded by uniform-rigidity spherical harmonic-domain approaches, which of course introduces some error in the elastic plate response when Te is varying. Assessments of inverse methods using synthetic data indicate that the errors are acceptably small for purposes of mapping Te variation using spatially-localized power-spectral methods, but the assumption of locally-uniform Te likely imposes limits on the resolution/bias properties of such methods. The modeling approach described here might be useful for both synthetic and inverse applications, if it were extended to permit both topographic and internal loading of the lithosphere, and to calculate the resulting gravitational perturbations.*

**A reference to Kirby (2022) has been added in Section 4 to provide information on commonly used methods for estimation of the effective elastic thickness.**

*With that said though, the application of the method to data from Europe (and elsewhere) has the potential to confuse unsophisticated readers (those who are not deeply-steeped in isostatic inversion methods). I can anticipate some readers believing that (e.g.) the Te or reference density estimates presented in Figure 11 have some bearing on reality, when really they are just parametric artefacts of imposing a single-input/single-output assumption on an inherently multiple-input/multiple-output problem (as first described in Forsyth, JGR 1985). The paper will be much more valuable to readers if that is clearly communicated in the abstract and conclusions, and if results given in §5 are caveated as a demonstration of application of the numerical technique, recognizing that additional enhancements to the modeling approach must be made (e.g., calculating gravity, incorporating additional seismic/density fields) before meaningful estimates of Te may result.*

**The main purpose of our paper is to introduce methods from isogeometric finite element analysis to the geoscientific community. Our intention is not to provide definitive results using a fully developed and geophysically accurate model of the lithosphere but to demonstrate the isogeometric framework for solving higher-order problems in geoscientific applications. It is the first step towards a useful numerical tool for the analysis of lithospheric deformations. This has been emphasized in the abstract and in the conclusion as suggested.**

*The additional comments below expand a bit on this key point, and I hope will be useful/-constructive in helping the authors to clarify that the results given in the paper are useful*

*for demonstrating the method rather than bearing on physical properties.*

*Line 18: "Globally $C^1$ finite element spaces..." The audience for EGUsphere is not mostly made up of mathematicians, to my knowledge, so it might be worthwhile to define terms like $C^1$ upon introduction...*

**A short description of $C^1$ is given in Appendix A: List of symbols. For clarity, we have added "continuously differentiable" in parentheses to the first mention of $C^1$.**

*Line 40: There has been significant change to European Moho maps since Grad et al. (2009), with the addition of hundreds (or thousands) of seismic stations and technical advances like ambient-noise Rayleigh wave imaging. Crust 1.0 (Laske et al. 2013) would be better, and even this has been improved on in more recent global models (e.g. SPiRaL, Simmons et al., GJI 2021). Given the limited utility of the simplified version of isostasy in the modeling performed in this paper, it has little impact: i.e., the parameter estimates derived in section 5 have little bearing on reality owing to oversimplification in the assumptions of load-forcing and response, for reasons described in the comment below regarding lines 163–165, so it doesn't matter whether the data used to constrain the model is the most accurate available. In future efforts though, assuming the method is extended to accommodate loading at the Moho, internal density variations and gravity prediction, it may be worth incorporating more recent estimates of data fields.*

**The suggestion to use more recent estimates of data for the numerical simulations is most appreciated. An extremely simplified model of the lithosphere, without the incorporation of many relevant processes, has been deliberately used in our work. The current implementation can be improved and extended in the future, so that the additional enhancements required for a useful model are taken into account.**

*Lines 53–56: It's worth being a bit careful in the wording here, as the notion of an "elastic lithosphere" is a red flag that can raise suspicions in the lithosphere community. The bending-stress moments that are supported by the nebulously-defined "lithosphere" are actually dynamically-maintained flow- and frictionally-supported stresses (hence the term "effective" in "effective elastic thickness", see e.g. McNutt & Menard JGR 1978, McNutt JGR 1984). The thin elastic plate approximation is useful solely because the bending moments supported in a thin elastic plate can be physically related to the bending moments supported in the real (dynamically-supported) lithosphere, and the two observationally appear the same for identical bending moments (e.g., Willett et al. Nature 1985; Burov et al. JGR 1995; Brown & Phillips JGR 2000). This enables valuable constraints to be placed on unknown rheological properties.*

**We reiterate that "elastic lithosphere" is meant according to the definition in Melosh (2011, Box 3.4), which is the notion considered in our work as opposed to the "real lithosphere". The sentence has been rephrased to emphasize that.**

*Lines 163–165: This treatment of the Moho as a surface of flexural displacement is problematic in this context. The Moho is a compositional boundary separating silica-rich ("crustal") from ultramafic ("mantle") rocks, and the thickness of the crust is determined by a combination of surface processes (erosion, deposition, volcanic extrusion, faulting), intrusive/magmatic processes (which are of order 10–50X greater volume than extrusions), tectonic strain, and lower-crustal flow. Each of these will have different transfer functions relating surface elevation to Moho deflection at flexural isostatic wavelengths, leading to*

*Forsyth's (JGR 1985) conceptualization of "surface" and internal or "subsurface" loads acting on the lithosphere. (The Moho can approximate a flexural deflection in response to treatment of topography as a load at very long wavelengths where the transfer functions approach those of Airy isostasy... But only if other buoyancy phenomena such as crustal density variations, thermal variations, or variable normal stress forcing by asthenospheric buoyancy are negligible contributors to vertical stress.) The practical consequence of this is that any estimate of Te arrived at by assuming the Moho is a flexural response to topographic loading will be meaningless for purposes of understanding long-term rheological support of stress by the lithosphere (i.e., the bending moment implied by the resulting estimate of Te will have no knowable relationship to the true bending moment). This is illustrated for example by comparing the map of Te in Europe (Figure 11a) with any well-tested inverse model incorporating a solution for surface and internal loading (e.g., Pérez-Gussinyé & Watts, Nature 2005), and by recognizing that the low Te's in Figure 11 are incompatible with a reasonable expectation of rheological controls on long-term stress-support in the region (consistent with the earlier finding by Forsyth, JGR 1985, that inversion using an assumption of topographic loading only yields extreme underestimates of Te).*

**We are aware that the flexural deflection will not necessarily directly correspond to the Moho depth variation and agree that inversion based on assumed topographic loading without the incorporation of subsurface variations is not adequate for identifying the effective elastic thickness, as illustrated by the poor result in Figure 11a. This is now addressed in Section 2.2 and 5.2. In future efforts, we will incorporate additional data such as gravity anomalies to provide more meaningful estimates of the effective elastic thickness using the isogeometric approach.**

*Figures (generally): Matlab has some (albeit limited) geographical mapping capabilities in its figure-making, and for those figures in which geographical regions are represented it would be very helpful to include (at the very least) the coastlines to help orient the reader. Flipping back and forth between the topography shown in Figure 10 and the maps in Figure 11 to orient oneself, for example, is cumbersome.*

**Coastlines have been included in the maps to better orient the reader. Filled contour plots are now used instead of colored contour lines to reduce visual clutter.**
* * *
Further changes to the manuscript:

**We have decided to completely remove Figures 9a to 9d from the manuscript to reduce the file size of the document due to the minor relevance and the requirement of 300 dpi for the resolution of the figure files. They contain visualizations of some results for the numerical experiments in the following geographic locations: the Java Island, the Indonesian Archipelago, the Hawaiian Islands, and the Himalayan Mountain Range, which can be retrieved from the software provided at `https://github.com/rozanxt/igalith` instead. If the editor deems it necessary to include the figures, we can revert the changes or publish them as supplementary material.**

---

## Author Response (AR2)

**Author's response to the reviews and relevant changes to the manuscript egusphere-2024-1093**

This document provides a detailed point-by-point response to all referee comments and the relevant changes that have been made in the manuscript. Referee comments are shown in *italics*. Author's responses are shown in **bold**.
* * *
Anonymous Referee #1:

*The authors have adequately addressed my remarks from the initial review. Thank you very much. Please address the following technical points before proceeding with final publication:*

*1. Issue has been addressed.*

*2. Issue has been addressed.*

*3. The issue has been mostly addressed. A generalized coordinate $\zeta$ has been introduced (see Eq. 3). Please make sure that all symbols provided in the paper are adequately defined.*

**All instances of the symbol $\zeta$ have already been replaced with the thickness parameter $\vartheta^3$ in the previous revision of the manuscript. The thickness parameter $\vartheta^3$ is now explicitly mentioned on page 4 of the revised manuscript. We have checked to the best of our ability that all symbols are defined adequately and available in Appendix A: List of symbols.**

*4. Thank you very much for the explanation. In the view of the additional information, I agree with the authors that straight two-dimensional beams can be thought of a one-dimensional simplification of the standard plate theory. The authors mentioned that this section is crucial for establishing a relation to the commonly used onedimensional model for flexural isostasy. Under these circumstances, I would encourage the authors to re-introduce this paragraph and make sure they adequately explain that the one-dimensional reduction does not introduce any other three-dimensional effects.*

**The paragraph has been re-introduced with an explanation on the absence of additional rotational effects for the beam model.**

*5. Thanks for the explanation, however this point is still confusing. In the revised form of the paper you write "Using the more general shell equations, it is possible to perform simulations of the lithosphere on the whole surface of the Earth." (page 10 in the revision). It seems that you spent considerable amount of space in the paper to present the equations for the plate, but you did not spend any effort in explaining the equilibrium equations for the shell. From a mathematical standpoint, the equilibrium equations for the shell are virtually an embedding of the plate equilibrium equations to a curvilinear space. However, this introduces significant mechanical effects and load carrying behaviors, that are not found in plates. It would be great if you could think of ways to unify your presentation in terms of plates and shells, because currently there is a deep presentation for the plate and almost no word spent for the shell.*

**The equilibrium equations for thin elastic shells are discussed in Section 2.1.1. They are reduced to the case of a plate with an initially flat configuration in Section 2.1.2. Topographic loading and buoyancy are introduced for the plate model in Section 2.2 and extended to the spherical shell model in Section 2.4.**

*6. Thanks a lot for the clarification.*

*7. Understood, thank you very much for the explanation.*

*8. Thanks for the explanation. I would use an italicized font when writing symbols attributed to functional spaces, such as Sobolev spaces.*

**Thank you for the suggestion. We would like to keep the upright font style for function spaces in our manuscript.**

*9. Thanks a lot for the explanation. If modeling the entire lithosphere using a single shell is an enormous simplication, which is clear, would it still be possible to use multipatch geometries to capture the physical reality in a more accurate manner? It might be worthwhile adding a note into your paper.*

**Multi-patch parametrizations are still useful for modeling domains with geometries that are more complex than quadrilaterals, e.g., the multi-patch example for Central Java. Simulations on a curved domain can provide results that are more accurate if the initial midsurface configuration of the lithosphere plays a significant role, which may be the case for certain locations or in large scales.**

*10. Issue has been addressed.*

*11. Issue has been addressed.*

*12. Issue has been addressed.*

*13. Issue has been addressed.*

*14. Issue has been addressed.*

*15. Issue has been addressed.*

*16.–17. Thanks a lot for the clarifications. Please consider adding this argumentation directly into your manuscript to make sure that ambiguity is removed.*

**Section 3.2.3 has already been modified in the previous revision of the paper to address most of the issues. A sentence has been rearranged and we have added the remark "Note that contiguous patches are assumed to share the same interpolatory control points at the interfaces to enforce the $\mathbf{C^0}$ continuity in our implementation."**

*18. Thanks a lot for the clarifications.*

*19. I think this enhanced my misunderstanding regarding the use of plates and shells in this work. I know understand, that on page 16 it is discussed yet plates in regard to the strong $C^1$ continuity, but in Section 5.3 it is used a shell for the example. Is then my understanding correct, that Section 3.2.3 discusses the $C^1$ (strong) multipatch coupling only in the setting of plates? I understand that for the shell (see example in Section 5.3) you are not using any multipatch coupling of shells, could you please clarify that somewhere in your text to remove ambiguity?*

**Section 3.2.3 discusses the strong multi-patch $\mathbf{C^1}$ coupling for both plates and shells, as indicated by the sentence "This has been extended from the case of planar multi-patch domains to multi-patch surfaces in Farahat et al. (2023a, b)." The numerical implementation used in our work is the one in Section 8.4 of Collin et al. (2016), which works for surfaces as well (see Section 6 therein) and has been used for the example discussed in Section 5.3 of our paper.**

*20. Thanks for clarifying. Please consider adding this information into your manuscript.*

**The phrase "when the data required for the simulation are only available on certain parts of Earth's surface" has been added to a relevant sentence on page 19 of the revision.**

*21. Thanks a lot for the clarification.*

*22. Thanks a lot for addressing the bug. This will make reproducibility of the results easier.*

**Thank you very much for testing our software and for all the other feedbacks.**
* * *
Referee #2 (Tony Lowry):

*With the changes, the paper is worthy of publication. For future work on this topic though, I would recommend that the authors read carefully and understand the concepts referred to earlier in papers by Forsyth (JGR 1985) and the textbook by Kirby (2022). Note particularly that estimation of subsurface loading is not as simple as inverting gravity, because both topography and gravity are functions of both flexure (hence the choice of flexural rigidity) and the loading distributions... Which places Te entirely within a null-space that can only be overcome by imposing statistical constraints on the problem.*

*Having spent a couple of decades now trying to advance the frontiers of seismic data analysis to improve upon estimates of the subsurface density structure (with a partial goal to improve estimates of Te), I can affirm with some confidence that the seismology community is still a very long distance from having sufficient confidence in seismically-constrained density structure to no longer need statistical (i.e., power spectral) approaches in estimating Te. So, when taking the next steps with this new approach, I hope you will also be prepared to incorporate power spectral estimation into the analyses!*

*Best regards,*
*Tony*

**Thank you very much for the helpful feedback and for suggesting acceptance of our paper for publication at Geoscientific Model Development. We will keep your remarks in mind and consider power spectral estimation in the analyses for future work on this topic.**
* * *
Further changes to the manuscript:

**Equations without numbering and equations that are made smaller to make space in the two-column version of the manuscript have been given numbers and returned to the default size in the one-column version, respectively. This should be fine-tuned in the final two-column version of the paper.**

**Missing gradient operators, denoted by the nabla symbol $\nabla$, in the definition of the $C^1$ constraint matrix (Equation 40 of Section 3.2.3 in the revised version) have been added. Some index placements (superscripts instead of subscripts) for the parameter $\vartheta$ in Section 3 have been fixed to distinguish coordinates from knot values. Some spacings have been added between the declaration and the definition of mappings in display-style math mode. A citation has been fixed: Hinze et al. (2009).**

---

## Author Response (AR3)

**Author's response to the reviews and relevant changes to the manuscript egusphere-2024-1093**

This document provides a detailed point-by-point response to all referee comments and the relevant changes that have been made in the manuscript. Referee comments are shown in *italics*. Author's responses are shown in **bold**.
* * *
Public justification by the topic editor:

*I am placing your paper as ready to publish pending technical corrections for two reasons:*

*(1) the two points given above by the GMD staff,*

**The full affiliations of authors have been added and the sections have been rearranged according to the manuscript composition as suggested by the GMD staff (now using the Copernicus Publications LaTeX Package, version 7.11, 9 April 2025).**

*(2) a note that you might consider the reviewer's clarifying question about thin elastic shells and whether further explanation, as suggested by the reviewer, might be helpful to potential readers.*

**We have re-evaluated the issue and assure that our exposition in the manuscript is sufficiently clear. Both the shell and plate models have been considered in our work. In contrast to the reviewer's standpoint, we start from the general formulation for thin elastic shells and consider its reduction to a plate model. This has been clarified in the previous response by pointing out the relevant sections. We believe that no further explanation is required in the manuscript.**